# Autoregressive-based Progressive Coding for Ultra-Low Bitrate Image Compression

**Ziyuan Zhang[1], Yichong Xia[1,5], Bin Chen[2,5], Tianwei Zhang[3], Hao Wang[4,6], Han Qiu[1]***

[1]Tsinghua University, China    [2]Harbin Institute of Technology, Shenzhen, China
[3]Nanyang Technological University, Singapore    [4]Peking University, China
[5]Peng Cheng Laboratory, China    [6]01.AI, China
`ziyuan-z23@mails.tsinghua.edu.cn`, `qiuhan@tsinghua.edu.cn`

## Abstract

Generative models have demonstrated significant results in ultra-low bitrate image compression, owing to their powerful capabilities for content generation and texture completion. Existing works primarily based on diffusion models still face challenges such as limited bitrate adaptability and high computational complexity for encoding and decoding. Inspired by the success of Visual AutoRegressive model (VAR), we introduce AutoRegressive-based Progressive Coding (`ARPC`) for ultra-low bitrate image compression, a progressive image compression framework based on next-scale prediction visual autoregressive model. Based on multi-scale residual vector quantizer, `ARPC` efficiently encodes the image into multi-scale discrete token maps and controls the bitrates by selecting different scales for transmission. For decompression, `ARPC` leverages the prior knowledge inherent in the visual autoregressive model to predict the unreceived scales, which is naturally the autoregressive generation process. To further increase the compression ratio, we target the VAR as a probability estimator for lossless entropy coding and propose group-masked bitwise multi-scale residual quantizer to adaptively allocate bits for different scales. Extensive experiments show that `ARPC` achieves state-of-the-art perceptual fidelity at ultra-low bitrates and high decompression efficiency compared with existing diffusion-based methods. All source code is available at `https://github.com/Joanna-0421/ARPC`.

## 1 Introduction

Neural image compression (Liu et al., 2023) has demonstrated superior performance than traditional methods (Wallace, 1991) for ultra-low bit compression tasks (Xia et al., 2025). Recently, generative models, such as GAN (Goodfellow et al., 2020) and diffusion model (Rombach et al., 2022), are explored for image compression at ultra-low bitrates and optimized for rate-distortion-perception (Blau & Michaeli, 2019), preserving texture details to improve human-perceptual performance.

Diffusion-based approaches have been proven to outperform GAN-based approaches for low bitrate image compression and achieve significant breakthroughs in the perceptual quality when decompressing. However, they still face three issues. 1) **Limited bitrate adaptability.** Most diffusion-based approaches (Careil et al., 2023; Li et al., 2024c; Xia et al., 2025) follow the *one-model-per-rate* training paradigm, which makes rate adaptation in real dynamic transmission environments difficult. 2) **High encoding and decoding complexity.** Although recent works (Theis et al., 2022; Vonderfecht & Liu, 2025) have explored progressive coding by combining pretrained diffusion models and reverse channel coding, the diffusion models' iterative denoising nature still introduces inevitable complexity. 3) **Infinite shared randomness**. The reverse channel coding requires that the sender and receiver must share randomness, which may not always be available (Lei et al., 2025).

In this paper, we aim to address above three issues via exploring using Visual AutoRegressive model (VAR) for image compression. VAR's next-scale prediction utilizes a multi-scale residual quantizer to encode an image into discrete, hierarchical visual tokens and later predict tokens progressively

---

*The corresponding author

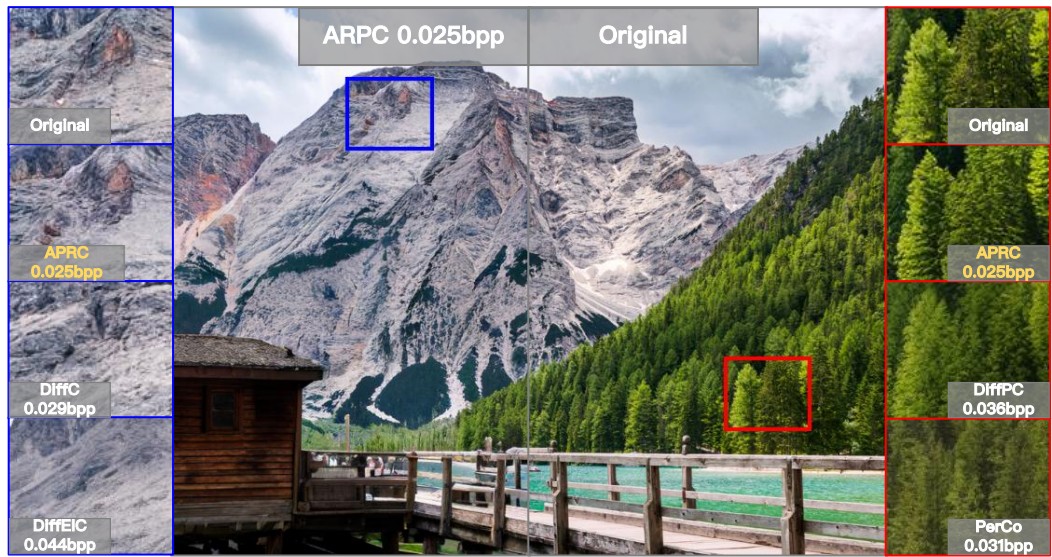

Figure 1: Qualitative comparison between `ARPC` and diffusion-based methods. `ARPC` effectively reconstructs fine-grained textural details, while other methods exhibit noticeable texture loss.

from coarse to fine in an autoregressive manner. Thus, our first key insight is that **this coarse-to-fine paradigm naturally aligns with a perfect bitrate adaptability** by transmitting coarse but crucial information first, such as layout, followed by progressively adding fine-grained texture details to gradually enhance image quality. Then, compared with diffusion-based approaches, VAR has two additional advantages including **faster generation process** and **no requirements for sharing randomness** between the sender and receiver.

Motivated by the above insights, we explore the potential of VAR for ultra-low bitrate image compression tasks, and propose **Auto**R**egressive-based **P**rogressive **C**oding (`ARPC`), an efficient progressive image compression framework for ultra-low bitrate image compression. When compression, `ARPC` utilizes the bitwise multi-scale residual quantizer to directly encode the images into $K$ bitstreams for transmission ($K$ is the predefined number of scales for residual maps) and controls the bitrates by only transmitting the first $k$ bitstreams. To further improve the compression ratio, we utilize lossless entropy coding to re-encode the original bitstream based on the probability predicted by VAR and propose the group-masked bitwise multi-scale residual quantizer (GM-BMSRQ), which further compresses the coarse information into fewer bits. For decoding, `ARPC` leverages VAR to generate other $K - k$ scale tokens in an autoregressive manner, naturally integrating the decompression task into the generative process of VAR, achieving progressive coding.

We compare `ARPC` with 13 state-of-the-art baselines on two commonly used image compression testsets, considering 6 perceptual metrics. Extensive experiments prove that `ARPC` significantly outperforms baselines across different metrics on all datasets at ultra-low bitrates. What's more, `ARPC` achieves higher decompression efficiency, 2∼6× faster than diffusion-based methods. Figure 1 illustrated that with the strong image prior of VAR, `ARPC` can achieve superior perceptual and statistical fidelity, effectively reconstructing fine-grained texture details with high realism.

## 2 RELATED WORKS

### 2.1 NEURAL IMAGE COMPRESSION METHODS

Learned image compression methods have achieved remarkable rate-distortion performance compared to traditional image compression standards like JPEG (Wallace, 1991). Some methods focus on the coding process, considering the probability estimation accuracy to increase the compression ratio, by introducing hyperprior entropy model (Ballé et al., 2018), autoregressive context modeling (Minnen et al., 2018), a discrete Gaussian mixture model (Cheng et al., 2020), and spatial-channel contextual

adaptive model (He et al., 2022a). Other methods focus on optimizing the model architecture to improve feature extraction ability through well-designed modules, such as mixed transformer-CNN (Liu et al., 2023) and frequency-aware transformer (Li et al., 2023a).

Although the above VAE-based methods have achieved good performance, optimizing solely for mean square error (MSE) can lead to excessive image smoothing and low statistical fidelity at low bitrates. (Blau & Michaeli, 2019) proposes to add a perceptual term in loss function to align well with human perception, which is used in many recent works (Mentzer et al., 2020; Muckley et al., 2023; He et al., 2022b), combining generative adversarial network and a perceptual loss term. Due to the strong prior of images and robust language understanding, text-to-image diffusion model (Rombach et al., 2022) has been explored for image compression tasks (Zhang et al., 2024; 2025b) and further improves the image fidelity and semantic consistency at ultra-low bitrates. (Lei et al., 2023) transmits image caption and image sketch as conditions for ControlNet (Zhang et al., 2023), enabling the image compression at utral-low bitrates. (Li et al., 2024c) and (Xia et al., 2025) utilize pre-trained stable diffusion to reconstruct images under the guidance of latent VAE features or image-level and semantic-level control flow. Although these methods significantly improve the image fidelity, they all suffer from long decoding time due to the large number of denoising steps. Recent works explore accelerating the decoding process by initiating denoising process from compressed latents (Li et al., 2025), dynamically aligning denoising steps with bitrates (Ke et al., 2025) and leveraging the distilled SD-Turbo prior for one-step reconstruction (Zhang et al., 2025a). All above methods need to train separate models for each bitrate, making rate adaptation difficult. (Theis et al., 2022) proposes DiffC to transmit the intermediate corrupted version of the diffusion process to achieve progressive compression based on reverse channel coding, assuming that shared randomness exists between the encoder and the decoder. (Vonderfecht & Liu, 2025) further improves the DiffC algorithm and gives the first complete implementation considering the computational complexity of reverse channel coding, but still suffers from long compression and decompression time.

## 2.2 DISCRETE VISUAL TOKENIZATION

VQVAE (Van Den Oord et al., 2017) first introduces the concept of discrete latent space, transforming the continuous representation into discrete vectors using a trainable codebook. With further development, such as LFQ (Yu et al., 2023), BSQ (Zhao et al., 2024), and residual vector quantization (Zeghidour et al., 2021), discrete visual tokenization has also been widely used for compression tasks, such as audio compression (Zeghidour et al., 2021), image compression (Careil et al., 2023; Mao et al., 2024; Guo et al., 2025), and video compression (Tian et al., 2023). By transmitting the indices of discrete vectors, the compression methods based on codebooks can not only control the upper bound of bitrates (Careil et al., 2023) but also avoid inconsistent probability distribution estimation problems in cross-platform scenarios (Tian et al., 2023). However, to ensure the low bitrate, the vocabulary size of the codebooks is often small, which can't cover all image features, resulting in suboptimal compression performance and low perceptual fidelity.

## 2.3 VISUAL AUTOREGRESSIVE MODELS

Driven by the great success of LLMs (Brown et al., 2020; Touvron et al., 2023), visual autoregressive models utilize discrete visual tokenizers, such as VQVAE (Van Den Oord et al., 2017), to convert image patches into index-wise tokens, generating images based on next-token prediction (Lee et al., 2022; Wang et al., 2024) or next-scale prediction (Tian et al., 2024; Han et al., 2025). Based on next-scale prediction, VAR (Tian et al., 2024) makes the autoregressive model surpass diffusion transformers (Peebles & Xie, 2023) for the first time, considering image quality, inference speed, and scalability. Infinity (Han et al., 2025) further proposes bitwise modeling using BSQ (Zhao et al., 2024), improving the scaling and visual detail representation capabilities of VAR. VAR has also been further expanded to other tasks, such as conditional generation (Li et al., 2024b), image restoration (Wang et al., 2025), and image super-resolution (Qu et al., 2025), demonstrating the potential of VAR for various tasks. However, to the best of our knowledge, the potential of VAR for image compression task remains unexplored.

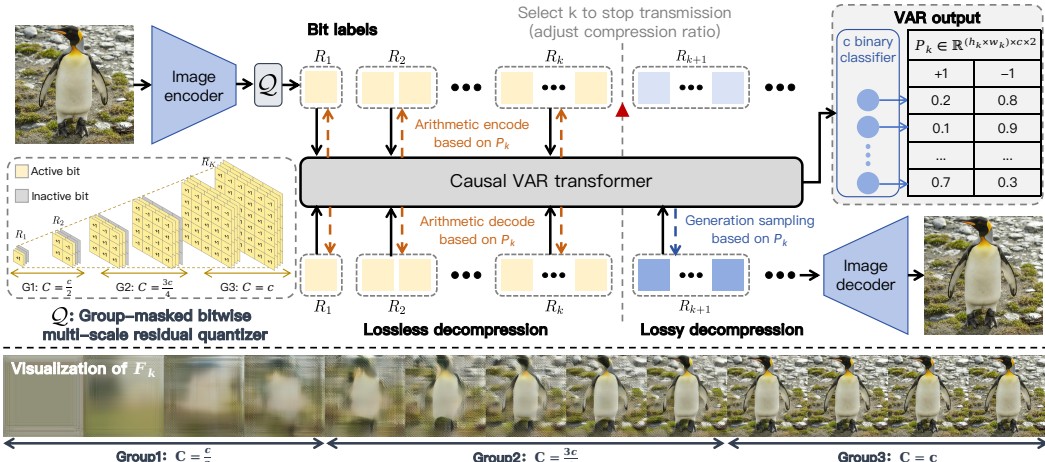

Figure 2: Overview of `ARPC`. `ARPC` leverages coarse-to-fine generation paradigm of VAR to achieve progressive compression (section 3.2). `ARPC` encodes the image into $K$ scale tokens and applies arithmetic coding for lossless compression based on the probability predicted by VAR (section 3.3). To further improve the compression ratio, `ARPC` utilizes group masks to enhance the original quantizer to compress image into fewer bits (section 3.4).

## 3 METHOD

In this section, we first introduce the preliminary of visual autoregressive model (section 3.1). Then, we give the detailed explanation of the `ARPC`'s framework to achieve the progressive compression and decompression (section 3.2). To improve the compression ratio, we use lossless entropy coding to re-encode the indices of the tokens (section 3.3). Based on the observation of visualization for different scale residual token maps, we propose group-masked bitwise multi-scale residual quantizer (GM-BMSRQ). This enhances the original quantizer by achieving a more compact representation at earlier scales, thus further improving the compression ratio at ultra-low bitrates (section 3.4).

### 3.1 PRELIMINARIES: VISUAL AUTOREGRESSIVE MODEL

We focus on the VAR based on next-scale prediction, where the basic unit of autoregression is the entire scale, containing a sequence of tokens. For an image $x$, the image encoder first encodes $x$ into a feature map $F \in \mathbb{R}^{h \times w \times c}$, and then quantize the feature map $F$ into $K$ scale residual maps, denoted as $(R_1, R_2, ..., R_K)$. The resolution of $R_k$, $(h_k \times w_k)$, grows larger gradually from $k = 1 \rightarrow K$. Each scale corresponds to $h_k \times w_k$ tokens. For simplicity, we use scale tokens to denote the sequence of tokens of each scale. We denote $F_k$ as the cumulative sum of the upsampled $R_{\leq k}$: $F_k = \sum_{i=1}^{k} upsample(R_i, (h, w))$. As $k$ increases, $F_k$ gradually approximates the continuous feature map $F$. Such quantization method ensures a coarse-to-fine paradigm, with earlier scales presenting the crucial structural information, while the subsequent scales progressively add finer details. We give the visualization of $F_k$ in Figure 2. Inspired by Infinity (Han et al., 2025), we use bitwise multi-scale residual quantizer, where $r_{i,j} = R_k^{(i,j)} = \frac{1}{\sqrt{c}} sign(\frac{r_{i,j}}{|r_{i,j}|})$.

In the discrete latent space, the VAR learns to predict the next scale tokens $R_{k+1}$ conditioned on previous scale tokens $R_{\leq k}$ and text input $t$, which can be formulated as:

$$p(R_1, R_2, ..., R_K) = \prod_{i=1}^{K} p(R_k | R_1, ..., R_{k-1}, C),$$

$(1)$

where $C$ is the text embedding, which is also projected into the beginning token <SOS>, and $p \in \mathbb{R}^{h_k \times w_k \times c \times 2}$ is predicted by $c$ binary classifiers, presenting the probability of next scale tokens.

## 3.2 PROGRESSIVE COMPRESSION WITH AUTOREGRESSIVE FRAMEWORK

The basic idea behind our method is based on the coarse-to-fine paradigm, which prioritizes the transmission of crucial data, the earlier scale tokens of the multi-scale residual quantizer. To achieve this, `ARPC` only transmits the first $k$ scale tokens and generates other $K - k$ scale tokens by VAR, leveraging the generation capacity to add finer details, reconstructing high realism images. Note that `ARPC` can achieve progressive coding through selecting different $k$ scales for transmission, which enables the single compression model to support multiple bitrates.

At the sender end, we encode an image $x$ based on bitwise multi-scale residual quantizer, as shown in Figure 2, $\mathcal{E}(x) = (R_1, R_2, ..., R_K)$. Unlike transmitting the index of codebook in token-based methods (Careil et al., 2023; Guo et al., 2025), bitwise quantizer maps each $c$-dimensional vector to a $c$-bit binary code (i.e. 1 for positive, 0 otherwise), which can naturally be considered as bitstreams. The $k$-th scale tokens, $R_k$, naively needs $h_k \times w_k \times c$ bits for transmission. What's more, we use BLIP2 model (Li et al., 2023b) to generate an image caption $t$ as a global semantic context, transmitted to the receiver end along with the first $k$ scale tokens.

At the receiver end, we get the first $k$ scale tokens $R_{\leq k}$, and use them as the prefixed content to predict $\hat{R}_{>k}$ through $K - k$ autoregressive processes. All $K$ scale tokens are upsampled into the same resolution and concatenated together, then input into the image decoder $\mathcal{D}$ to get the reconstructed image $x_k = \mathcal{D}((R_1, ...R_k, \hat{R}_{k+1}, ..., \hat{R}_K))$.

To get a better reconstruction quality, we analyze the distortion of the reconstructed image $x_k$ ($k$ denotes only the first $k$ scale tokens transmission between sender and receiver) and demonstrate that the distortion $D_k = d(x, x_k)$ has an upper bound:

**Theorem 3.1** Given an input image $x$, encoder $\mathcal{E}$, decoder $\mathcal{D}$. Let $(R_{\leq k}, R_{>k}) = \mathcal{E}(x)$ be the ground truth tokens and $x_K = \mathcal{D}((R_{\leq k}, R_{>k}))$ be the reconstructed image from ground-truth tokens, with distortion $D_K = d(x, x_K)$. Let $\hat{R}_{>k}$ be the tokens predicted by VAR based on $R_{\leq k}$, and $x_k = \mathcal{D}((R_{\leq k}, \hat{R}_{>k}))$ be the final reconstruction, with distortion $D_k = d(x, x_k)$. We have:

$$\mathbb{E}\left[D_k\right] \leq \mathbb{E}\left[D_K\right] + C \cdot \mathbb{E}_{R_{\leq k}}[D_{KL}\left(p\left(R_{>k}|R_{\leq k}\right)\|p_\theta\left(R_{>k}|R_{\leq k}\right)\right)]. \tag{2}$$

See the proof in Appendix A.1. It indicates the distortion can be optimized by minimizing two key components: 1) the reconstruction error between the original image and the image decoded from ground-truth tokens, and 2) the KL divergence between the true and predicted distributions of the scale tokens. Thus, the training has two stages. First, we train image encoder, decoder, and quantizer to minimize the first distortion term in Equation 2, following:

$$\mathcal{L}_{first} = \mathcal{L}_{rec} + \mathcal{L}_{per} + \mathcal{L}_{dis} + \mathcal{L}_{commit} + \mathcal{L}_{entropy}, \tag{3}$$

where $\mathcal{L}_{rec}$ is the $\mathcal{L}_1$ loss between reconstructed image $x_K$ and input image $x$, $\mathcal{L}_{per}$ is the perceptual loss, $\mathcal{L}_{dis}$ is the discriminator loss, $\mathcal{L}_{commit}$ is MSE loss in the discrete latent space and $\mathcal{L}_{entropy}$ is the entropy loss used for bitwise quantizer (Zhao et al., 2024). Second, we optimize the VAR to minimize the KL divergence between the true and predicted distribution, following:

$$\mathcal{L}_{VAR} = -\sum_{i=1}^{K} \log p_\theta(R_i|R_{<i}). \tag{4}$$

## 3.3 LOSSLESS RE-ENCODING FOR DISCRETE INDICES

To further improve the compression ratio, (Careil et al., 2023) assumes that the token indices satisfy a uniform distribution for lossless entropy coding. Different from the vector quantizer, bitwise quantizer index each token using a binary code, denoted as $y_k(i, j) = \sum_{n=0}^{c-1} \mathbb{1}_{R_k(i,j,n)>0} 2^n$, where $i, j$ is the spatial coordinate and $n$ is the channel coordinate of the $k$ scale tokens. However, each element of $y_k(i, j)$ doesn't satisfy the uniform distribution (i.e., 0.5 for 1 and 0.5 for 0), as they represent semantic information of the image and are therefore dependent on the previous context. Interestingly, we notice that the learning objective of the VAR is to accurately predict such distribution conditioned on previous scales, as illustrated in section 3.1. This provides the potential for more effective entropy coding, as the distributions are highly accurate.

At the sender end, we target the VAR as a probability estimator, which uses $c$ binary classifiers to predict $c$ independent distributions for each token, denoted as $p_{k(i,j)} \in \mathbb{R}^{c \times 2}$. We use arithmetic

coding (the details can be found in Appendix A.3), a lossless entropy coding, to further compress the token indices based on $p_k$. For compression, we follow the autoregressive process. First, we use the beginning token <SOS> to predict the probability of the first scale, $p_1$, to encode the indices $y_1$. Then, the bit indices of each scale can be encoded by the probability predicted VAR conditioned on previous scales until all $k$ scale indices are encoded in an autoregressive manner. The decompression can be divided into two stages: the lossless decompression and lossy decompression. The lossless decompression is the same as the compression process, predicting the probability of each scale tokens from the first to the $k$-th, and using arithmetic decoding for lossless decompression. For lossy decompression, we leverage the generation capacity of VAR and use classifier free guidance on predicted logits to generate $K - k$ scale tokens to get the final decompressed image.

## 3.4 GROUP-MASKED BITWISE MULTI-SCALE RESIDUAL QUANTIZER

ARPC achieves ultra-low bitrate compression by transmitting only the first few scale tokens. The first few scale tokens contain the image's most crucial coarse-grained information, such as the image structure and color. Although crucial, they are low-frequency information with less detail, which can be compressed into a smaller feature space. Bitwise multi-scale residual quantizer downsamples the feature maps from spatial dimension to create less tokens. Inspired by the product quantization (Guo et al., 2024), which divides a high-dimensional vector into several low-dimensional tokens using different quantizers, we propose to enhance the bitwise multi-scale residual quantizer with group masks, compressing the feature space of the first few scales considering both resolution and channel dimension, to further increase the compression ratio at ultra-low bitrates.

First, we observe each scale, noting that they have a clear hierarchy and can be broadly divided into three groups, as shown in Figure 2. The first four scales present large color blocks, establishing the macro color layout of the image. The subsequent five scales begin to show the basic outlines and structure of objects, but still lack details, resulting in the blurriness of edges. The last four scales significantly improve the fine textures of the images. Based on our observation, we divide $K$ scales into three groups and propose group-masked bitwise multi-scale residual quantizer (GM-BMSRQ), masking the channels of scale tokens. For a feature map $F \in \mathbb{R}^{h \times w \times c}$, with the original scale tokens $R_k \in \mathbb{R}^{h_k \times w_k \times c}$, we mask the last $\frac{c}{2}$ channels for the first group, setting to -1 as inactive bits, and the last $\frac{c}{4}$ for the second group. Such method adaptively creates a more compact representation of information based on the content of each scale, while further reducing the number of bits required for the transmission of the first two groups.

Besides reducing the token bit number, a smaller value of $k$ leads to a lower bitrate. However, (Li et al., 2024a) reveals that the image semantics are concentrated in the final few scale tokens, which means that if we only transmit the first few scale tokens, the receiver will get little useful information for decompression, resulting in low reconstructed image quality at low bitrates. To solve this problem, we apply scale random dropout (SRD) strategy during training to enhance the semantic information preservation in the earlier scale tokens, retaining more semantic information at ultra-low bitrates.

## 4 EXPERIMENTS

### 4.1 EXPERIMENT SETUP

**Datasets.** We use Coyo-700M (Byeon et al., 2022), a large-scale image-text pair dataset, as our training dataset. We curated our training dataset through a multi-stage pipeline. We first select images with a resolution greater than $1024 \times 1024$, and then exploit an OCR model to filter undesired images with too much text. Finally, we utilize the advanced InternVL 2.0 model (Gao et al., 2024) to re-caption all filtered images to provide more accurate and detailed annotations. Our final training dataset includes 5M highly curated images with detailed captions.

For evaluation, we validate ARPC on two widely used image compression benchmark datasets: DIV2K-val (Agustsson & Timofte, 2017) and CLIC2020 (Toderici et al., 2020), which consist of 100 and 428 high-definition images, with the captions generated by BLIP2 model. We reference (Xia et al., 2025; Guo et al., 2025) to center-crop all original images to $1024 \times 1024$ resolution for evaluation. As Kodak lacks extensive validation, we present the experimental results in Appendix A.4.

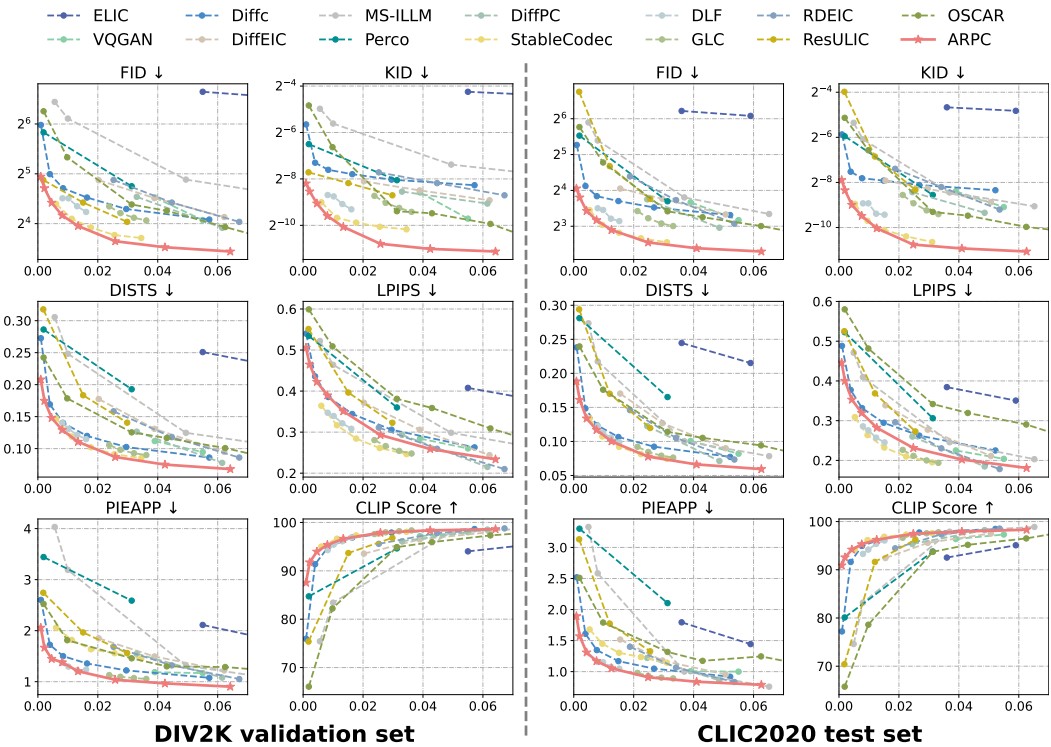

Figure 3: Quantitative comparisons with SOTA methods on the DIV2K and CLIC2020 datasets.

**Model training details.** The training has two stages. First, we train the image encoder and decoder with the group-masked bitwise multi-scale residual quantizer, which is configured with three groups using channel dimensions of 8, 12, and 16, respectively. The training begins with 500k iterations on $256 \times 256$ images, followed by an additional 300k iterations on a mix of $256 \times 256$, $512 \times 512$, and $1024 \times 1024$ resolutions. The scale random dropout strategy is used with a probability of 0.2 and applied from the fourth scale. In the second stage, we freeze the image encoder and decoder, use Infinity-2B as the visual autoregressive model, and finetune it for 20k iterations using the AdamW optimizer with a batch size of 64 and a learning rate of $6 \times 10^{-5}$ on $1024 \times 1024$ images with text captioned by InternVL 2.0 model. All experiments are performed on $8 \times$ NVIDIA H20 GPUs.

**Metrics.** We focus more on perceptual quality at ultra-low bitrates following previous works (Xia et al., 2025; Vonderfecht & Liu, 2025) and select 6 perceptual metrics for quantitative measures. For perceptual similarity comparison, we consider LPIPS (Zhang et al., 2018) and DISTS (Ding et al., 2020). For human-centric visual quality comparison, we consider PIEAPP (Prashnani et al., 2018), focusing on human preference for perceptual fidelity, and CLIP Score (Hessel et al., 2021), focusing on high-level semantic content comparison. For overall distribution comparison, we consider FID (Heusel et al., 2017) and KID (Bińkowski et al., 2018), which capture the statistical fidelity between compressed and original images.

**Baselines.** We compare ARPC with several state-of-the-art neural compression schemes, including VAE-based, diffusion-based, and token-based methods. For VAE-based methods, we consider ELIC (He et al., 2022a), optimizing for rate-distortion and MS-ILLM (Muckley et al., 2023), optimizing for rate-distortion-perception. For diffusion-based methods, we consider DiffEIC (Li et al., 2024c), DiffPC (Xia et al., 2025), RDEIC Li et al. (2025), ResULIC Ke et al. (2025), StableCodec Zhang et al. (2025a) and OSCAR Guo et al. (2025), which are all based on a pre-trained diffusion model to reconstruct images conditioned on the decoded content variables. Furthermore, we consider DiffC (Vonderfecht & Liu, 2025), which is also a progressive compression method by transmitting the corrupted version of diffusion process. We use the DiffC based on the pretrained Flux-dev model. Token-based methods include VQGAN (Mao et al., 2024), GLC Qi et al. (2025) and DLF Xue et al.

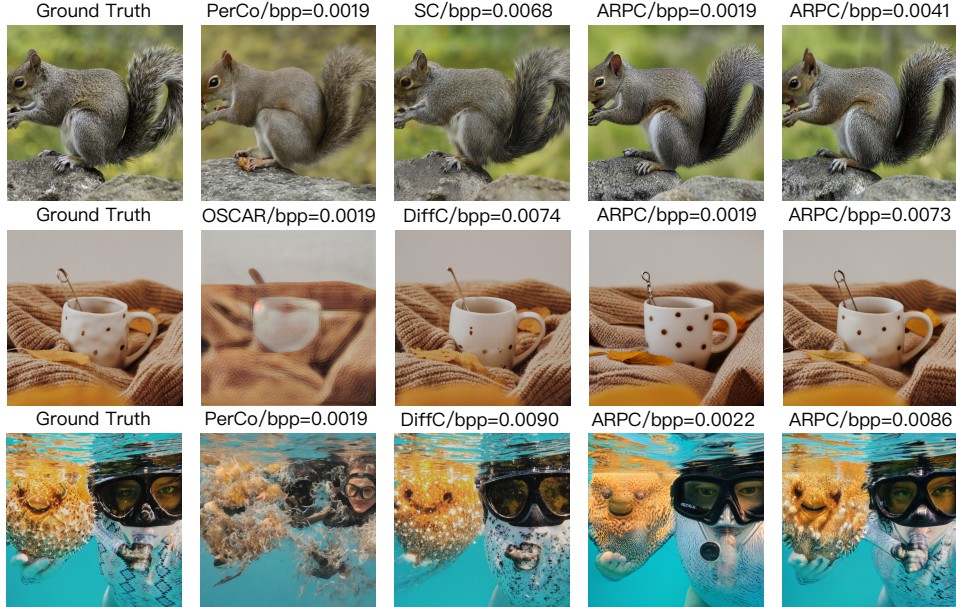

Figure 4: Visual comparison of generative compression methods at ultra-low bitrates (<0.01bpp). SC denotes the StableCodec.

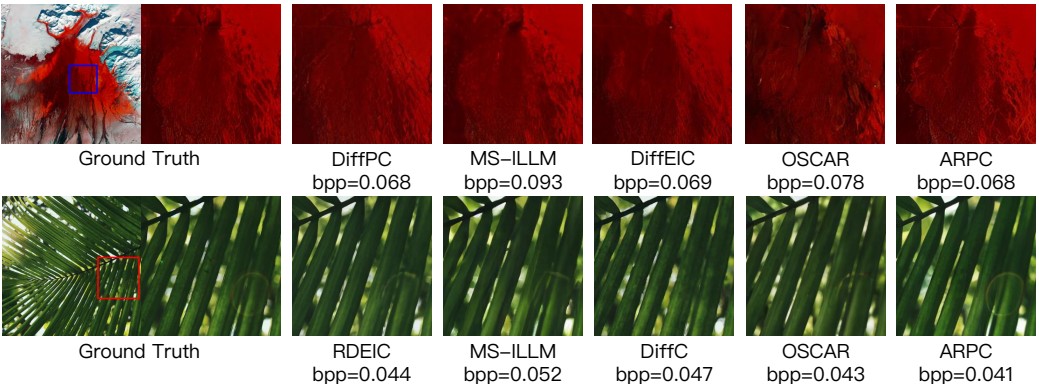

Figure 5: Qualitative illustrations of various methods on DIV2K and CLIC2020 datasets.

(2025). For all baselines, we retrain their models with our training dataset using their open-source code and default settings, achieving performance consistent with the original results.

## 4.2 MAIN RESULTS

**Quantitative comparisons.** Figure 3 show the comparison results on DIV2K validation set and CLIC2020 test set. The results demonstrate that ARPC consistently outperforms most baseline solutions across all transmission rates, especially at ultra-low bitrates. Furthermore, ARPC significantly outperforms all baselines as measured by the FID and KID metrics, demonstrating our approach's excellent capacity for preserving statistical fidelity. We attribute this superiority over diffusion-based methods to the powerful generative capabilities inherent in autoregressive-based architecture. Compared to DiffC, a diffusion-based progressive codec, ARPC outperforms it across all bitrates without the need for shared randomness due to latent feature discretization. Compared to token-based methods, which also utilize discrete token indices for transmission, ARPC achieves significantly better performance. This is due to two main factors: ARPC utilizes a larger codebook that covers more comprehensive image features, controlling bitrates by transmitting different numbers of scales,

| Methods | Steps | Encoding (s) | Decoding (s) | BD-rate(%) | | |
|---|---|---|---|---|---|---|
| | | | | FID | DISTS | PIEAPP |
| PerCo (Careil et al., 2023) | 20 | 0.20 | 10.25 | 1167.35 | 882.09 | 2744.75 |
| DiffEIC (Li et al., 2024c) | 50 | 0.65 | 15.98 | 681.76 | 139.64 | 100.75 |
| DiffPC (Xia et al., 2025) | 50 | 0.17 | 23.66 | 93.90 | 20.49 | 16.52 |
| RDEIC (Li et al., 2025) | 6 | 0.31 | 4.18 | 523.52 | 1469.67 | 761.27 |
| StableCodec (Zhang et al., 2025a) | 1 | 0.42 | 1.11 | 0.1547 | 6.99 | 254.42 |
| DiffC (Vonderfecht & Liu, 2025) | - | $3.63 \sim 45.22$ | $13.63 \sim 37.25$ | 674.37 | 59.62 | 117.11 |
| ARPC | 13 | 1.8~6.2 | **5.39** | 0 | 0 | 0 |
| ARPC (w/o LRE) | 13 | 0.20 | **5.39** | 34.64 | 34.58 | 34.38 |

Table 1: Inference efficiency and BD-rate (%) comparison on CLIC2020 dataset (1024×1024).

but not the vocabulary size of the codebook. What's more, ARPC leverages VAR to more accurately predict the distribution of token indices, further increasing the compression ratio.

**Qualitative comparisons.** We provide qualitative comparisons in Figure 4, which shows the visual results at ultra-low bitrates (<0.01 bpp), and Figure 5, which shows the image detail comparison at higher bitrates. To ensure low bitrates, PerCo and OSCAR uses a vocabulary size of 256, which can only cover simple image features. As illustrated in Figure 4, for a simple image, such as a squirrel, PerCo can preserve the basic shape and semantics, but fails to reconstruct the original colors. For more complex images, however, PerCo and OSCAR are unable to retain the original semantic information. DiffC and StableCodec successfully reconstructs the basic structure and semantic content of the image, but it suffers from a loss of fine details, such as the face of the human or the pin in the cup. ARPC maintains a higher similarity in structure, semantics, and details, resulting in significant visual improvements. At higher bitrates, ARPC demonstrates a superior capability in reconstructing high-frequency details. As shown in Figure 5, competing methods like MS-ILLM and OSCAR exhibit various degrees of blurring or structural distortion. In contrast, ARPC successfully preserves fine-grained textures, such as the delicate patterns and the speckles of light on leaves, demonstrating the remarkably superior visual quality and realism of ARPC with minimal bitrates.

**Computational complexity compared with diffusion-based methods.** VAR achieves high generation efficiency by adopting a hierarchical generation strategy with a small number of iterations. Thus, ARPC significantly reduces the decoding latency compared to the diffusion-based methods, which need longer inference time due to multi-step denoising process. As shown in Table 1, ARPC only requires 5.39s to decompress an image, which is twice as fast as PerCo and 2~6× faster than DiffC. Compared with one-step diffusion based methods, such as StableCodec, although ARPC requires more decoding time, the most significant advantage of ARPC is the progressive coding, which can not be achieved by the one-step diffusion based method. Different from DiffC, which can initiate generation from an arbitrary denoising step, AR-based model predicts the $k + 1$-th scale depending on the first $k$ scales. Consequently, ARPC starts decoding process from the beginning token and completes a predefined number of steps to decode an image, resulting in a deterministic decoding time rather than a variable range. For encoding, due to ARPC targets the VAR as a probability predictor, transmitting $k$ scales tokens requires predicting the probability distribution of the first $k$ scale tokens for arithmetic coding. Thus, the encoding time is variable, increasing with higher bitrates. Unlike diffusion-based methods, which exhibit a consistent computational cost per iteration, the complexity of ARPC's encoding process is very low at low bitrates with fewer token counts at early scales (refer to Appendix A.7 for more details).

## 4.3 ABLATION STUDY

In this section, we conduct the ablation study by removing **LRE (lossless re-encoding)**, **SRD (scale random dropout)**, and **GM-BMSRQ (group-masked bitwise multi-scale residual quantizer)**, and the quantitative results are shown in Figure 7. Since both the LRE and the group mask for the BMSRQ are used to improve the compression ratio with little impact on the decompressed image quality, we only present the visual results without SRD, as illustrated in Figure 6.

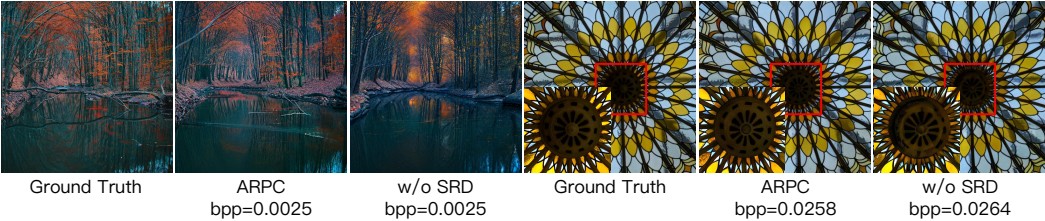

| Ground Truth | ARPC
bpp=0.0025 | w/o SRD
bpp=0.0025 | Ground Truth | ARPC
bpp=0.0258 | w/o SRD
bpp=0.0264 |

Figure 6: Visual comparison with and without scale random dropout strategy.

**W/o LRE.** To demonstrate that the lossless entropy re-encoding based on the probability predicted by the VAR can efficiently reduce the bitrates, we remove the arithmetic coding and directly transmit the binary code quantized by the GM-BMSRQ. As shown in Figure 7, a bitrate reduction of approximately 30% can be achieved by applying arithmetic coding, without any impact on image quality. Without the computational overhead of autoregressive probability prediction, the encoding time is significantly reduced, requiring only 0.2 seconds, as shown in Table 1.

**W/o SRD.** To show that scale random dropout strategy can enhance the representation capability of earlier scales, improving the fidelity of reconstructed images at low bitrates. We remove such strategy during the first stage training, and results are shown in Figure 7. The results show that SRD plays an important role in decompressing images to get better perceptual metrics, especially at low bitrates. As illustrated in Figure 6, under 0.01 bpp, SRD is crucial for maintaining the fundamental structure and color information. At higher bitrates, SRD can still help to preserve fine-grained details, such as the patterns on the roof.

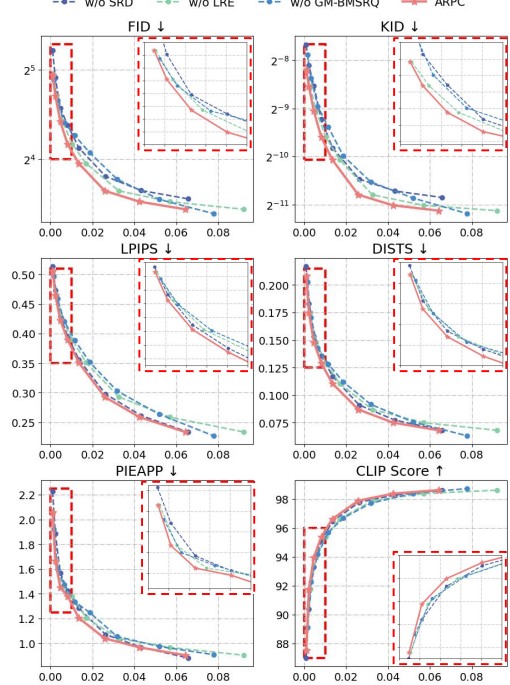

Figure 7: Ablation study on the DIV2K dataset.

**W/o GM-BMSRQ.** To prove the group mask used for BMSRQ can improve the compression ratio at ultra-low bitrates, we train the BMSRQ with the channel dimension of 16 without using group mask. As shown in Figure 7, removing the group mask leads to a noticeable rise in bitrates, demonstrating that group mask effectively improves the compression ratio, especially at ultra-low bitrates.

## 5 CONCLUSION

We introduce a novel progressive neural compression framework, ARPC, which intuitively utilizes the coarse-to-fine generation paradigm of VAR based on next-scale prediction to achieve progressive coding, and leverages the priors of the pre-trained VAR to reconstruct images with high realism and visual quality at ultra-low bitrates. Extensive results indicate that ARPC can not only achieve SOTA progressive compression performance but also improve the decompression efficiency. We discussion the LLMs usage in Appendix A.9.

## ACKNOWLEDGEMENT.

This work was supported by the National Natural Science Foundation of China (62576122, 62301189), and the Center of High performance computing, Tsinghua University.

ETHICS STATEMENT

All the researches in this paper conform the ICLR Code of Ethics[1]. Our efforts include, but are not limited to: 1) The method we propose can be used for progressive image compression systems, which meets social needs, and can be broadly accessible, without threats to health, safety, personal security, and privacy. 2) All the datasets, pretrained models and baseline methods are used in ways consistent with their licences, with full respect given to their copyright and contributions. 3) Our experiments are conducted with the best efforts to avoid potential biases. This includes using the same training dataset, testing with the same settings, and a comprehensive set of metrics for evaluation. 4) Transparency and reproducibility are prioritized. We will release the source code and model checkpoints upon acceptance, and the detailed description to reproduce our method is given in the manuscript.

REPRODUCIBILITY STATEMENT

Experiments are described in detail in section 4.1, including the pretrained model, the dataset, all the hyperparameters, and the model training details. The source code and model checkpoint will be released upon acceptance with detailed instructions including data access and preparation, the exact command and environment needed.

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

## A APPENDIX

### A.1 PROOFS

**Theorem 3.1** Given an input image $x$, encoder $\mathcal{E}$, decoder $\mathcal{D}$. Let $(R_{\leq k}, R_{>k}) = \mathcal{E}(x)$ be the ground truth tokens and $x_K = \mathcal{D}((R_{\leq k}, R_{>k}))$ be the reconstructed image from ground-truth tokens, with distortion $D_K = d(x, x_K)$. Let $\hat{R}_{>k}$ be the tokens predicted by VAR based on $R_{\leq k}$, and $x_k = \mathcal{D}((R_{\leq k}, \hat{R}_{>k}))$ be the final reconstruction, with distortion $D_k = d(x, x_k)$. We have:

$$\mathbb{E}[D_k] \leq \mathbb{E}[D_K] + C \cdot \mathbb{E}_{R_{\leq k}}[D_{KL}(p(R_{>k}|R_{\leq k}) \| p_\theta(R_{>k}|R_{\leq k}))] \tag{5}$$

*Proof.* According to the triangle inequality:

$$d(x, x_k) \leq d(x, x_K) + d(x_K, x_k) \tag{6}$$

$$\to D_k \leq D_K + d\left(\mathcal{D}((R_{\leq k}, R_{>k})), \mathcal{D}\left((R_{\leq k}, \hat{R}_{>k})\right)\right) \tag{7}$$

Assuming that the decoder $\mathcal{D}$ satisfies the Lipschitz continuity:

$$d\left(\mathcal{D}((R_{\leq k}, R_{>k})), \mathcal{D}\left((R_{\leq k}, \hat{R}_{>k})\right)\right) \leq L \cdot \delta\left(R_{>k}, \hat{R}_{>k}\right) \tag{8}$$

$$\mathbb{E}[D_k] \leq \mathbb{E}[D_K] + L \cdot \mathbb{E}\left[\delta\left(R_{>k}, \hat{R}_{>k}\right)\right] \tag{9}$$

$$= \mathbb{E}[D_K] + L \cdot \mathbb{E}_{R_{\leq k}}\left[\sum_{R_{>k}} p(R_{>k}|R_{\leq k})\left(\sum_{\hat{R}_{>k}} p_\theta\left(\hat{R}_{>k}|R_{\leq k}\right)\delta\left(R_{>k}, \hat{R}_{>k}\right)\right)\right] \tag{10}$$

$$\leq \mathbb{E}[D_K] + C \cdot \mathbb{E}_{R_{\leq k}}[D_{KL}(p(R_{>k}|R_{\leq k}) \| p_\theta(R_{>k}|R_{\leq k}))] \tag{11}$$

### A.2 MORE DETAILS OF ALGORITHM PROCEDURE

We supplement the explanation of the encoding and decoding processes of `ARPC` through pseudocode. The detailed encoding and decoding processes are shown in algorithm 1 and algorithm 2.

---

**Algorithm 1:** Encoding process of `ARPC`.

**input** : Input image $x$, image caption $t$
**output** : Encoded bitstream
**Parameters:** Image encoder with group-masked bitwise multi-scale residual quantizer $\mathcal{E}$, arithmetic codec $A$, select the first $k$ scales for transmission.

1 $(R_1, R_2, ..., R_K) = \mathcal{E}(x)$;
2 $< SOS > =$ text projection $(t)$;
3 $Bitstream_1 = A(R_1, p_\theta(R_1| < SOS >, t))$;
4 **for** $i=2,..,k$ **do**
5 $\quad | \quad Bitstream_i = A(R_i, p_\theta(R_i|R_1, ..., R_{i-1}, < SOS >, t))$;
6 **end**
7 transmission $Bitstream_1, ..., Bitstream_k$ and image caption $t$;

---

### A.3 ARITHMETIC CODING

Arithmetic coding is a form of lossless entropy coding. The core idea of arithmetic coding is to map an entire message to a single floating-point number within the interval $[0, 1)$. Based on the probabilities of each symbol, the current interval is partitioned into multiple sub-intervals. Through an iterative process, a very small interval is obtained, and its lower bound is used as the encoded result. The pseudocode of arithmetic coding is shown in algorithm 3 and algorithm 4.

---

**Algorithm 2:** Decoding process of `ARPC`.

---

**input** : $Bitstream_1, ..., Bitstream_k$, image caption $t$
**output** : Reconstructed image $x_k$
**Parameters:** Image decoder $\mathcal{D}$, arithmetic codec $A$

1   $<SOS>$ = text projection $(t)$;
2   $Bitstream_1 = A(R_1, p_\theta(R_1| <SOS>, t))$;
   /* Lossless decompression                                   */
3   $R_1 = A(Bitstream1, p_\theta(R_1| <SOS>, t))$;
4   **for** $i=2,..,k$ **do**
5     |   $R_i = A(Bitstream_i, p_\theta(R_i|R_1, ..., R_{i-1}, <SOS>, t))$;
6   **end**
   /* Lossy decompression                                           */
7   **for** $i=k+1,...,K$ **do**
8     |   $\hat{R}_i = Sampling(p_\theta(R_i|R_1, ..., R_k, \hat{R}_{k+1}, ...\hat{R}_{i-1}, <SOS>, t))$;
9   **end**
10 $x_k = \mathcal{D}((R_1, ..., R_k, \hat{R}_{k+1}, ..., \hat{R}_K))$;

---

---

**Algorithm 3:** Arithmetic Encoder

---

**Input:** A binary sequence $S = (s_1, s_2, \ldots, s_L)$, the probability of symbol '0', denoted as $p_0$
**Output:** The encoded value $v$

   // Initialize the interval [low, high) to [0, 1)
1   $low \leftarrow 0.0$;
2   $range \leftarrow 1.0$;

3   **for** *each symbol s in S* **do**
4     |   **if** $s = $ *'0'* **then**
5     |     |   $range \leftarrow range \cdot p_0$;
               // Narrow the interval for '0'
6     |   **else**
7     |     |   $low \leftarrow low + range \cdot p_0$;
               // Shift the interval for '1'
8     |     |   $range \leftarrow range \cdot (1 - p_0)$;
               // Narrow the shifted interval

9   **return** $low$;

---

### A.4   FURTHER EXPERIMENT RESULTS

We further consider the distortion metrics, PSNR and MS-SSIM, for comparison. The results are shown in Figure 8 and Figure 9. The last point on the rate-distortion curve represents the case where all $K$ scales of tokens are transmitted from the sender to the receiver. In this scenario, the receiver only takes the VAR as a probability estimator for lossless decompression, not for generation. Therefore, the compression performance at this bitrate is equivalent to the VAE's reconstruction performance. Since the reconstruction performance of a discrete VAE is worse than that of a continuous VAE, `ARPC`'s compression performance shows a slight drop compared to DiffPC at higher bitrates. However, it still achieves SOTA results on FID and KID, demonstrating `ARPC`'s superior statistical fidelity.

Furthermore, we present the comparison results on Kodak dataset, as shown in Figure 10. We don't report the FID and KID score due to the limited size of Kodak dataset. The results show that `ARPC` maintains outstanding performance on the Kodak dataset as well. We provide the qualitative comparisons in Figure 11. The results demonstrate that `ARPC` can reconstruct images with complex and correct textures, such as the color around the eye.

---

**Algorithm 4:** Arithmetic Decoder

---

**Input:** The encoded value $v$, the probability of symbol '0', $p_0$, the length of the original
            sequence, $L$
**Output:** The decoded sequence $S_{out}$

---

1  $low \leftarrow 0.0$;
2  $range \leftarrow 1.0$;
3  Initialize an empty sequence $S_{out}$;

4  **for** $i \leftarrow 1$ **to** $L$ **do**
5     $threshold \leftarrow low + range \cdot p_0$;
      // Calculate the split point
6     **if** $v < threshold$ **then**
7        $s \leftarrow$ '0';
8        $range \leftarrow range \cdot p_0$;
         // Select the interval for '0'
9     **else**
10       $s \leftarrow$ '1';
11       $low \leftarrow threshold$;
12       $range \leftarrow range \cdot (1 - p_0)$;
         // Select the interval for '1'
13    Append $s$ to $S_{out}$;

14 **return** $S_{out}$;

---

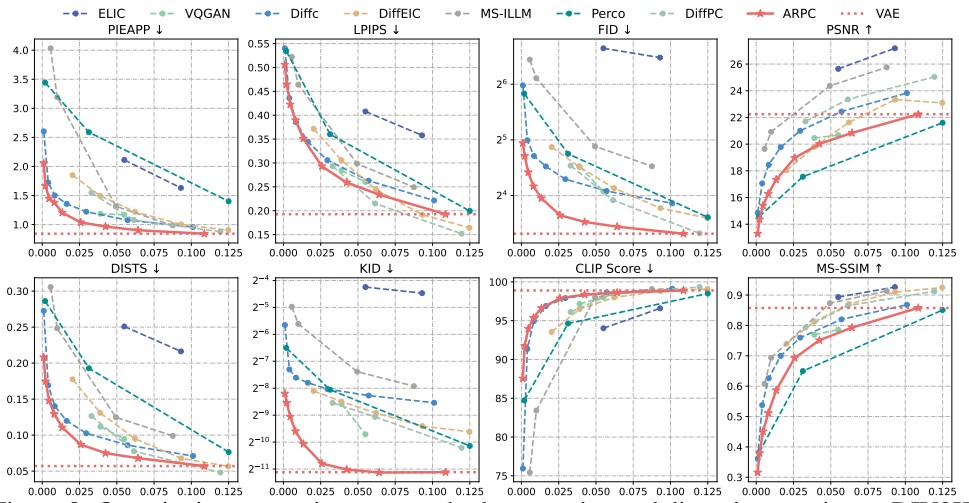

Figure 8: Quantitative comparisons across both perception and distortion metrics on DIV2K.

## A.5 FURTHER ABLATION EXPERIMENT

### A.5.1 VOCABULARY SIZE OF BITWISE MULTI-SCALE RESIDUAL QUANTIZER

Improving the vocabulary size of the bitwise multi-scale residual quantizer can improve the reconstruction capability of the VAE, but increase the bitrates at the same time. We further explore the effect of increasing the vocabulary size on compression performance. We increase the channel dimension of the bitwise multi-scale residual quantizer to $c = 32$, and configure the channels for the three groups to be 16, 24, and 32, respectively. As shown in Figure 13, at low bitrates, the decompressed image quality is similar for both models. However, the model with a channel dimension of 32 can achieve higher bitrates and exhibits better performance at higher bitrates. This is mainly due to the bitwise VAE has a better reconstruction performance with a larger vocabulary size.

### A.5.2 ABLATION RESULTS OF BITWISE VISUAL TOKENIZER

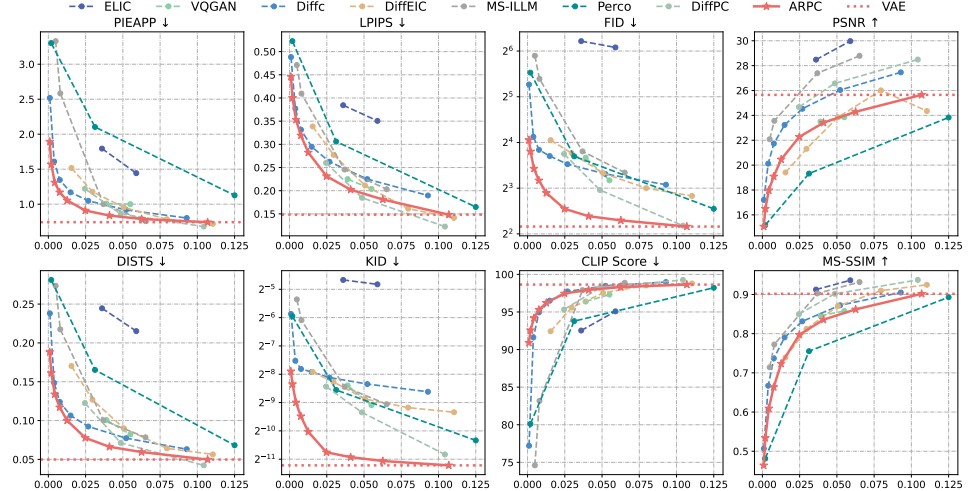

Figure 9: Quantitative comparisons across both perception and distortion metrics on CLIC2020.

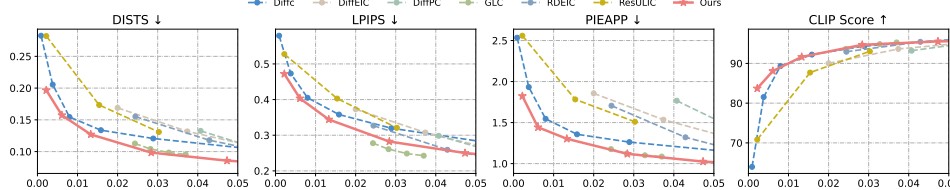

Figure 10: Quantitative comparisons with SOTA methods on Kodak dataset.

We divide $K$ scales into three groups based on the observation of each scale and apply different group mask for each group, retaining the bitwise visual tokenizer to adaptively allocate bits for different scales. What's more, we apply scale random dropout strategy to enhance the semantic information preservation in the earlier scales. We show the effect of the above two strategies on reconstruction quality of bitwise visual tokenizer in Table 2. We use the validation set of ImageNet for evaluation with the input resolution of $512\times512$. Since these strategies are all designed to improve the compression performance at ultra-low bitrates, either by reducing the vocabulary size of earlier scales or by focusing on earlier scales to retain more semantic information, they inevitably have a slight impact on the overall reconstruction capability of the VAE.

| Method | IN-512 rFID | IN-512 PSNR |
|---|---|---|
| Original | 0.31 | 22.6 |
| W/ GM-BMSRQ | 0.56 | 22.12 |
| W/ GM-BMSRQ W/ SRD | 0.67 | 21.5 |

Table 2: The effect of GM-BMSRQ and SRD on visual tokenizer.

### A.5.3 ABLATION STUDY OF IMAGE CAPTION LENGTH

We use the BLIP2 model to generate the image caption in this paper. And we further evaluate if a more detailed image caption can bring better compression performance, as illustrated in Figure 14. We use DIV2K validation set as our testset and use the advanced InternVL 2.0 model to provide more detailed captions. The results show that captions generated by BLIP2 and InterVL 2.0 yield similar compression performance. For CLIP Score, using the caption from InternVL 2.0 model can achieve higher score at ultra-low bitrates. This is mainly because the CLIP Score measures semantic similarity, and more detailed captions can provide more detailed semantics for the generation process. What's more, we evaluate the compression performance without image caption. As shown in Figure 14, at ultra-low bitrates, the absence of an image caption has a severe impact on the quality of the decompressed image. The tokens of earlier scales primarily encode structural layout, lacking finer semantic details of the original images, making the decompression heavily dependent on the guidance of a caption. As the number of transmitted scales increases, at higher bitrates, comparable compression performance is achievable even without a caption.

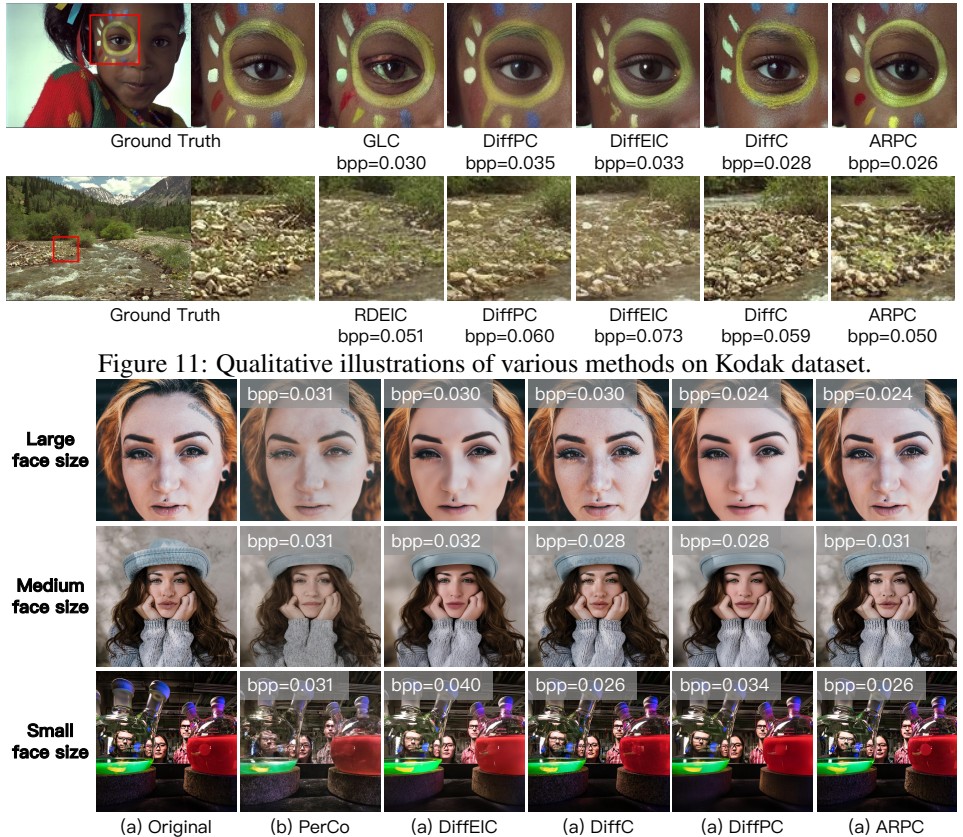

Figure 11: Qualitative illustrations of various methods on Kodak dataset.

Figure 12: The image containing human face.

### A.5.4 ABLATION STUDY OF DIFFERENT MASK CONFIGURATION

Based on our observation, we divide 13 scales into three groups of sizes 4, 5, and 4, respectively. We further give two different choices for masking, dividing 13 scales into (5,6,2) and (2,2,9). Table 3 shows the BD-rates on DIV2K dataset with the default configuration (i.e., 4,5,4) as the baseline. The results show that configuration 1, (5,6,2), compresses more information into a low-dimensional feature space, which can achieve lower bitrates. However, it severely compromises the representation capability of GM-BMSRQ, resulting in a degradation in the reconstructed image. While configuration 2, (2,2,9), effectively preserves the representation capability, it leads to a significant reduction in the compression ratio, especially at ultra-low bitrates.

| Configuration | BD-rate (%) | |
| --- | --- | --- |
| | DISTS | FID |
| (4,5,4) | 0 | 0 |
| (5,6,2) | 25.69 | 64.19 |
| (2,2,9) | 17.29 | 29.87 |

Table 3: BD-rate of different mask configuration for GM-BMSRQ.

### A.5.5 COMPARISON WITH VANILLA VAR

To demonstrate the superiority of ARPC over the vanilla VAR, we report the compression performance of directly using VAR for encoding and decoding, and compared with the SOTA compression method, StableCodec, as shown in Table 4. The BD-rate is calculated on DIV2K dataset with ARPC as the baseline. In vanilla VAR, the image semantics are concentrated in the final few scale tokens, which is suboptimal for progressive compression. Moreover, the earlier scales represent coarse, low-frequency information with low information density, but they share the same high-dimensional feature space and code-

| Methods | BD-rate (%) | |
| --- | --- | --- |
| | DISTS | FID |
| Vanilla VAR | 63.32 | 79.51 |
| StableCodec | 10.78 | 22.47 |
| ARPC | 0 | 0 |

Table 4: BD-rate comparison with vanilla VAR.

book as later scales with high information density. This results in representation redundancy, leading to inefficient information transmission, particularly at ultra-low bitrates.

| $k$ | 1 | 2 | 3 | 4 | 5 | 6 | 7 | 8 | 9 | 10 | 11 | 12 | 13 |
|---|---|---|---|---|---|---|---|---|---|---|---|---|---|
| Encoding time (s) | 1.86 | 1.90 | 1.94 | 1.99 | 2.03 | 2.07 | 2.15 | 2.27 | 2.46 | 2.77 | 3.35 | 4.34 | 6.21 |
| BPP | 7.3e-6 | 3.4e-5 | 1.4e-4 | 3.8e-4 | 9.6e-4 | 2.2e-3 | 4.6e-3 | 8.1e-3 | 0.013 | 0.025 | 0.042 | 0.064 | 0.108 |

Table 5: The encoding time and bitrate when selecting different $k$.

| Image size | Encoding (s) | Decoding (s) |
|---|---|---|
| $256 \times 256$ | $0.09 \sim 0.45$ | 0.44 |
| $512 \times 512$ | $0.28 \sim 1.39$ | 1.33 |
| $512 \times 768$ | $0.46 \sim 2.40$ | 2.29 |
| $1024 \times 1024$ | $1.86 \sim 6.21$ | 5.39 |

Table 6: Inference efficiency with different input image size.

## A.6 VISUAL RESULTS FOR SPECIAL SCENARIOS

We further present visual results for some challenging scenarios, such as the images containing face and text. We categorized text and faces into three classes following DiffPC based on their proportions within the entire frame - large, medium and small, as shown in Figure 12 and Figure 15. For images containing face, `ARPC` can reconstruct high realism images even at low bitrates. While other diffusion-based methods exhibit significant artifacts or texture loss. PerCo shows noticeable color incorrect. DiffEIC and DiffPC present oversmoothing of the skin for large and medium faces. For images containing text, `ARPC` can also achieve good image reconstruction quality, even for the small text, such as "TESLA.COM" under the license plate.

## A.7 SELECTION OF $k$

### A.7.1 COMPRESSION EFFICIENCY

As we illustrated in section 4.2, `ARPC` utilized the probability predicted by VAR to losslessly re-encode the token indices. As a result, the encoding time increases with higher bitrates, a large value of $k$. However, unlike the diffc, which has consistent computational cost per iteration, the encoding complexity is low at low bitrates as fewer token counts at early scales. We further provide the specific encoding times and bitrates for different numbers of transmitted scales $k$. As shown in Table 5, when transmitting tokens from fewer than 10 scales, the encoding time is under 3 seconds and the bitrate is below 0.025 bpp. As the number of tokens in the last 3 scales increases, it leads to a significant increase in compression time. Furthermore, we provide the encoding and decoding times for different input image sizes, as shown in Table 6, demonstrating the low decoding complexity of `ARPC`.

### A.7.2 VISUAL RESULTS OF THE PROGRESSIVE DECOMPRESSION.

We further present the visual results of transmitting different scales from $k = 5$ to $k = 13$ (all discrete tokens are transmitted), as shown in Figure 16. For a smaller $k$ (low bitrates), the decompressed image is similar to the original image only in terms of semantics and image layout. Therefore `ARPC` relies more on the VAR's generative capacity to reconstruct a high-realism image. As the value of k increases, more tokens are transmitted, and the details of the image are gradually completed.

## A.8 LIGHTWEIGHT DEPLOYMENT FOR COMPUTATIONAL RESOURCES-CONSTRAINED SCENARIOS

To demonstrate the potential for practical applications, we conduct further experiments using BLIP (a lightweight caption model) and consider a lightweight deployment strategy. We give the parameter counts and BD-rate comparison in Table 7 and Table 8.

- Replacing BLIP-2 with BLIP: We replaced the large BLIP-2 model with the more lightweight BLIP model. As shown in the BD-rate table below, utilizing BLIP (w/ LRE) yields compression performance very similar to the baseline BLIP-2.

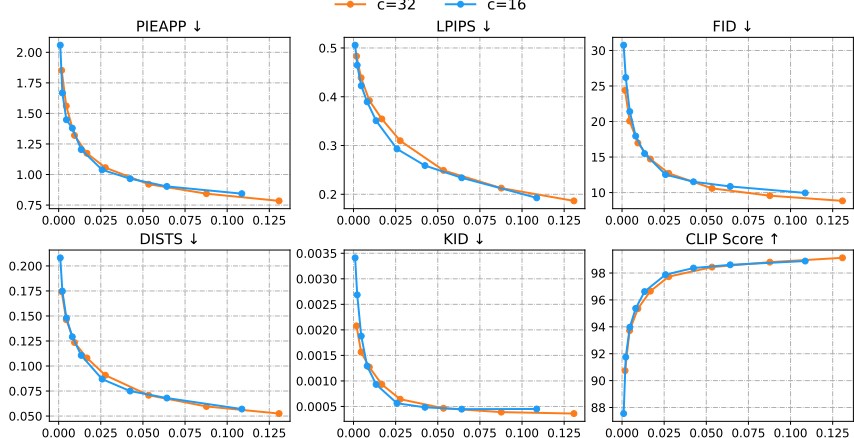

Figure 13: Quantitative ablation study of vocabulary size for group-masked bitwise multi-scale residual quantizer. Tested on DIV2K validation set.

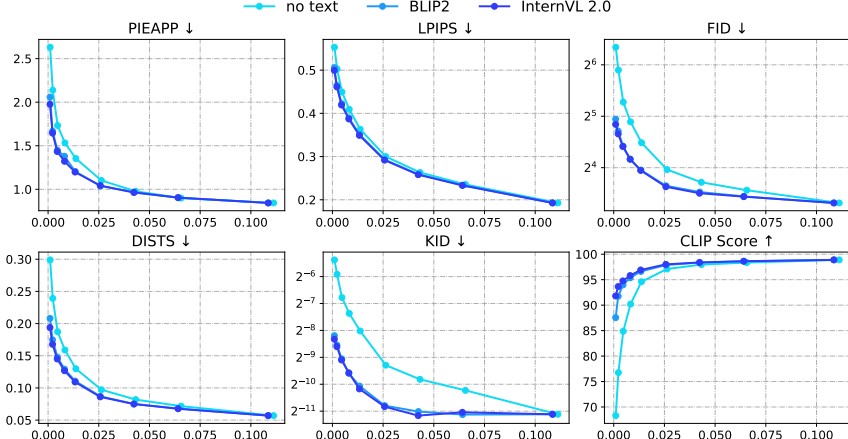

Figure 14: Quantitative ablation study of image captions with different length.

- Lightweight deployment w/o LRE: For scenarios strictly constrained by computational resources at the sender's end (e.g., mobile devices), we can directly transmit token indices without lossless re-encoding (LRE). This eliminates the need to deploy the VAR model at the sender, requiring only the VAE encoder and the lightweight caption model. Such strategy reduces the parameters at the sender's end from 2.2B to 268.8M. Although removing LRE results in a BD-rate increase ($\sim 25\%$) due to the lack of probability estimation, it offers a viable solution for edge devices.

Table 7: Model parameters for lightweight deployment using BLIP and w/o LRE.

| Model | Sender Params | Receiver Params |
|---|---|---|
| BLIP | 223.9M | 0 |
| VAR (w/o VAE) | 2.2B | 2.2B |
| VAE | 44.9M | 65.0M |
| Total (w/ LRE) | 2.2B | 2.2B |
| Total (w/o LRE) | 268.8M | 2.2B |

Table 8: BD-rate comparison for lightweight deployment using BLIP and w/o LRE. The BD-rate is calculated on DIV2K dataset with using BLIP-2 caption model and LRE as the baseline.

| Methods | BD-rate (%) | |
|---|---|---|
| | DISTS | FID |
| BLIP-2 + w/ LRE | 0 | 0 |
| BLIP + w/ LRE | 0.12 | 0.08 |
| BLIP + w/o LRE | 25.93 | 25.56 |

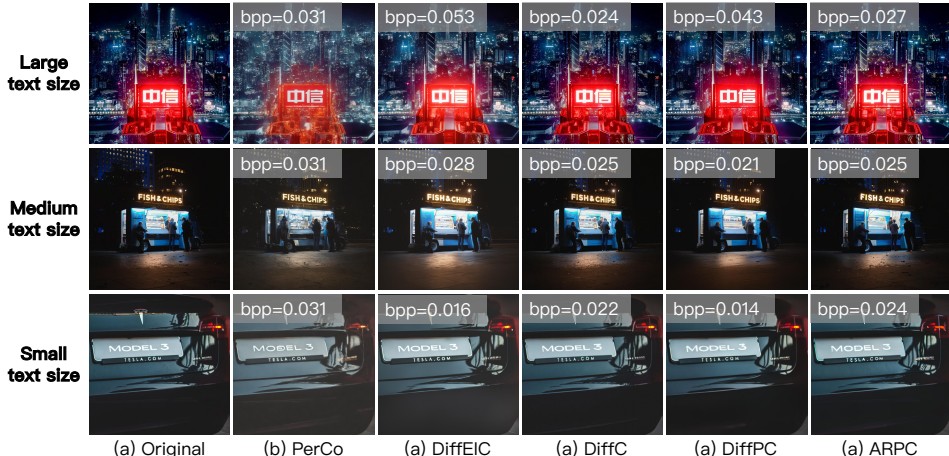

Figure 15: The image containing text.

## A.9 THE USE OF LARGE LANGUAGE MODELS

In this paper, we only use the LLMs for grammar checking and writing refinement as a general-purpose assist tool. All scientific content, including methodologies, data, and conclusions, is proposed by the authors and not influenced by the use of LLMs. All usage of LLMs is conducted with careful oversight by the authors to ensure accuracy.

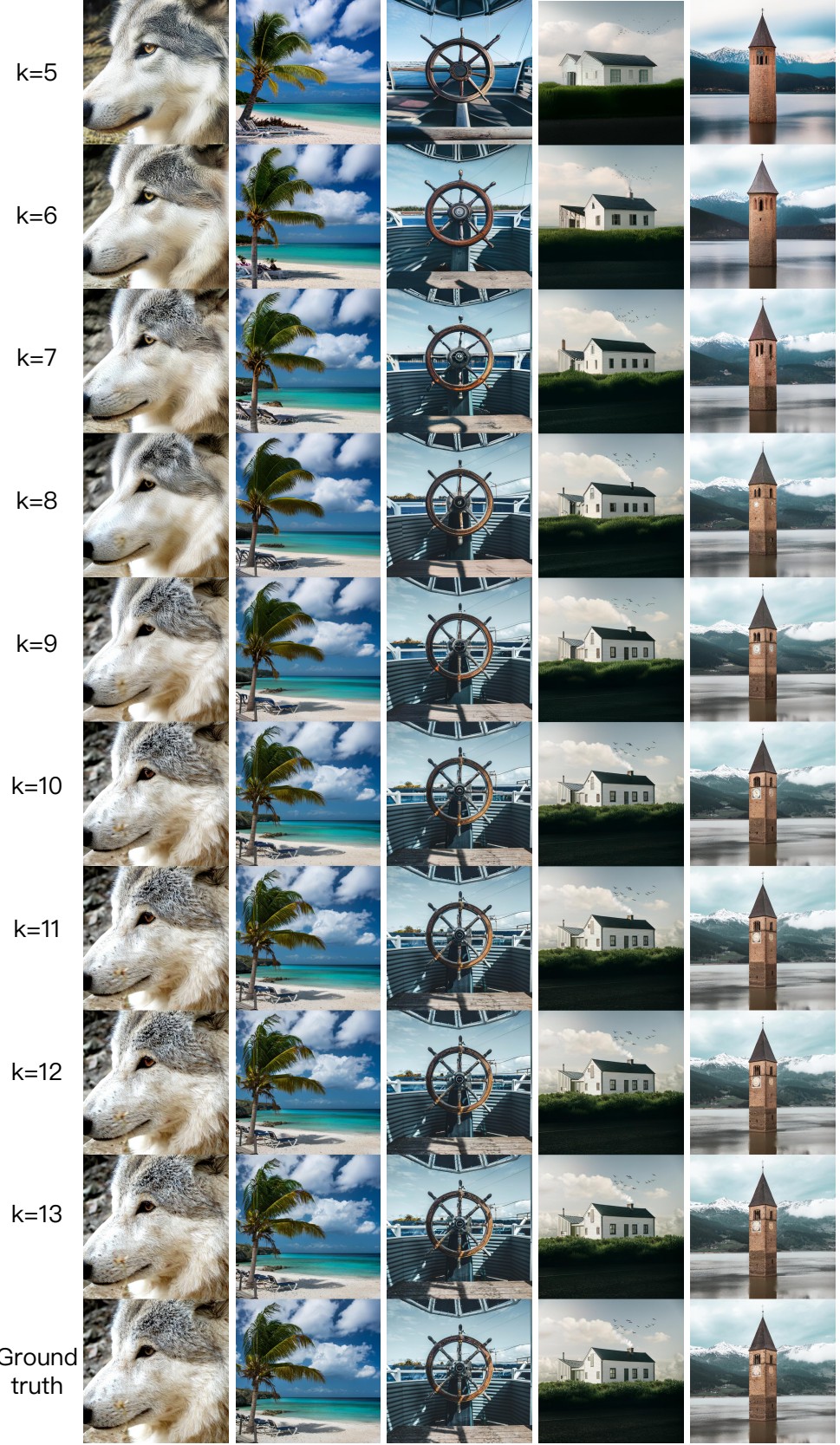

Figure 16: Visual comparison for different value of $k$.

