# OpenReview forum: "Autoregressive-based Progressive Coding for Ultra-Low Bitrate Image Compression"
_ICLR.cc/2026/Conference — ICLR 2026 Poster_

### Official Review · Reviewer_t8Sb · 2025-10-28

**Soundness:** 3
**Presentation:** 3
**Contribution:** 3
**Rating:** 6
**Confidence:** 5

**Summary:**

This paper proposes a novel framework for ultra-low bitrate image compression , which utilizes a Visual AutoRegressive model (VAR) to encode images into multi-scale discrete tokens. Decompression is achieved by autoregressively generating (predicting) the unreceived scales.

**Strengths:**

This paper achieves variable bitrate by combining VAR with lossless compression across different scales, effectively integrating image compression with existing pre-trained models.

The paper is well-organized and easy to understand.

**Weaknesses:**

1. The encoding time increases with the bitrate. Additionally, the encoding process requires a caption model to generate captions, making the encoder relatively "heavy." Could the authors provide the parameter counts for the models used at both the encoder and decoder ends?

2. The ablation study in Figure 7 shows that while each proposed module contributes some improvement, the individual gains are relatively modest. Could it be that directly using VAR for encoding and decoding already offers a performance advantage over other generative compression methods?

3. Further experiments are needed to investigate the impact of different specific mask choices for the Group-Masked (GM-BMSRQ) method.

**Questions:**

see weakness. My main point of interest is whether VAR itself is already sufficiently effective when used as a codec, and what the performance improvement of this paper's method is compared to a baseline VAR.

---

> ### Author Response · Authors · 2025-11-23
> **Author response (part 1/3)**
>
> Dear Reviewer t8Sb, thank you very much for your careful review of our paper and insightful comments. We are encouraged by your positive comments on **effective integration of VAR with lossless compression for variable bitrate**, and the **clear organization and readability of the paper**. We hope the following responses can alleviate your concerns and clarify key points.
>
> ---
>
>
> **W1:** The encoding time increases with the bitrate. Additionally, the encoding process requires a caption model to generate captions, making the encoder relatively "heavy." Could the authors provide the parameter counts for the models used at both the encoder and decoder ends?
>
> **R1:** We fully understand your concerns, and we address them from two aspects:
> * **Encoding time:** We respectfully clarify that the time increase with bitrate is an inherent characteristic of progressive coding rather than a design flaw. The traditional progressive compression methods, such as SVC [1], also exhibit encoding time increasing with bitrates. This is because the progressive codec gradually adds details to achieve higher bitrate compression. Therefore, higher quality requires more computational steps. Progressive coding also offers advantages that fixed-rate codecs lack: the receiver can stop receiving at any step to decompress a complete image without waiting for the entire bitstream to be received, which is critical for bandwidth-constrained scenarios.
> * **Model parameters:**  We acknowledge that the caption model introduces additional parameters. However, utilizing text as a global semantic prior is becoming a standard paradigm in diffusion-based compression (e.g., PerCo [2], DiffC [3], and DiffPC [4]) to enhance reconstruction fidelity at ultra-low bitrates. We provide the parameter counts for ARPC used at both the encoder and decoder ends:
>
> | Model  | Sender Params | Receiver Params |
> | ------ | ------------- | --------------- |
> | BLIP-2 | 4.2B          | 0  |
> | VAR |    2.2B       |   2.2B     |
> | Total | 6.4B        | 2.2B     |
>
> What’s more, we give the parameter counts for DiffC, another progressive coding method, for comparison.
>
> | Model  | Sender Params | Receiver Params |
> | ------ | ------------- | --------------- |
> | DiffC | 12B          | 12B  |
>
> What’s more, we also provide the ablation study without image caption in Appendix A.5.3. The results show that at ultra-low bitrates, the caption is crucial for semantic preservation. At higher bitrates, comparable performance can be achievable without the image caption.
> Furthermore, we also notice that BLIP-2 is large and brings more computational cost. We try to use BLIP, a lightweight model, to generate the image caption. And the compression performance is very similar to BLIP-2, indicating that users can select lightweight caption models to reduce the encoding cost with minimal impact on quality.
>
> ---
>
> **Reference**
>
> [1] https://en.wikipedia.org/wiki/Scalable_Video_Coding.
>
> [2] Careil M, Muckley M J, Verbeek J, et al. Towards image compression with perfect realism at ultra-low bitrates[C].ICLR2024.
>
> [3] Vonderfecht J, Liu F. Lossy compression with pretrained diffusion models[J].ICLR2025.
>
> [4] Xia Y, Zhou Y, Wang J, et al. DiffPC: Diffusion-based High Perceptual Fidelity Image Compression with Semantic Refinement[C].ICLR2025.

---

> ### Author Response · Authors · 2025-11-23
> **Author response (part 2/3)**
>
> **W2:** The ablation study in Figure 7 shows that while each proposed module contributes some improvement, the individual gains are relatively modest. Could it be that directly using VAR for encoding and decoding already offers a performance advantage over other generative compression methods?
>
>
>
> **R2:** We appreciate the reviewer's crucial question regarding the effectiveness of the vanilla VAR as a codec and the performance improvement of ARPC. To alleviate your concerns, we report the compression performance of directly using VAR for encoding and decoding, and compared with the SOTA compression method, StableCodec, as shown in the table below. The BD-rate is calculated on DIV2K dataset with ARPC as the baseline. These experiments have also been added to Appendix A.5.5.
>
>
> | Method | BD-rate (%) - DISTS | BD-rate (%) - FID |
> | ------ | ------------------- | ----------------- |
> | StableCodec | 10.78 | 22.47|
> | Vanilla VAR | 63.32| 79.51 |
> | ARPC | 0 | 0|
>
> Further, we clarify our contributions and the performance gap as follows:
>
> - ARPC is the first work to adapt multi-scale autoregressive generation model for image compression. We bridge the gap between generation and compression.
> - **The challenges to directly use vanilla VAR as a progressive codec**
>     - **Inefficiency at Ultra-low Bitrates**: In vanilla VAR, the image semantics are concentrated in the final few scale tokens, which is suboptimal for progressive compression. An effective progressive codec requires the earlier scales to contain as much compact information as possible. Moreover, the earlier scales represent coarse, low-frequency information with low information density, but they share the same high-dimensional feature space and codebook as later scales with high information density. This results in representation redundancy, leading to inefficient information transmission, particularly at ultra-low bitrates.
>     - **Lack of entropy optimization:** As VAR is based on discrete tokens for generation, a naive adaptation of VAR for compression is to transmit the indices of tokens, just like previous token-based compression methods. However, such strategy ignores the strong spatial and cross-scale correlations inherent in images.
> - To address the challenges above, ARPC has three improvements:
>     - **Scale random dropout:** To solve the issues that the image semantics are concentrated in the final few scale tokens, we apply scale random dropout during training.
>     - **Group mask mechanism:** To mitigate the redundancy caused by sharing the same dimensional feature space for both low information density and high information density scales, ARPC proposes to mask different channels for earlier scales, further compacting the information density.
>     - **Entropy coding:** To overcome the inefficiency of naively transmitting the token ID, ARPC dual uses the VAR transformer, fully exploring the VAR transformer as a probability estimator. ARPC utilizes the autoregressive logits predicted by the VAR transformer to estimate the probability distribution of the next scale tokens, combined with arithmetic coding to further increase the compression ratio without introducing a new module.

---

> ### Author Response · Authors · 2025-11-23
> **Author response (part 3/3)**
>
> **W3:** Further experiments are needed to investigate the impact of different specific mask choices for the Group-Masked (GM-BMSRQ) method.
>
> **R3:** We appreciate the suggestion. In this paper, based on our observation, we divide 13 scales into three groups of sizes 4, 5, and 4, respectively. We further give two different choices for masking, and provide the BD-rate in the table below. The BD-rate is calculated on DIV2K dataset with the default configuration as the baseline. These experiments have also been added to Appendix A.5.4.
>
> - **Configuration 1 (5, 6, 2)**: We choose a more aggressive strategy, applying masking to a larger proportion of scales. This configuration compresses more information into a low-dimensional feature space, which can achieve lower bitrates. However, it severely compromises the representation capability of GM-BMSRQ, resulting in a degradation in the reconstructed image.
> - **Configuration 2 (2, 2, 9)**: We mask only the earlier few scales and keep most scales with the original high-dimensional feature space. Although this configuration effectively preserves the representation capability, it leads to a significant reduction in the compression ratio, especially at ultra-low bitrates.
>
> | Configuration | BD-rate (%) - DISTS | BD-rate (%) - FID |
> | ------ | ------------------- | ----------------- |
> | Default configuration (4,5,4) | 0 | 0|
> | Configuration 1 (5, 6, 2) | 25.69|64.19|
> | Configuration 2 (2, 2, 9) | 17.29 |29.87|

---

> > ### Comment · Reviewer_t8Sb · 2025-11-25
> >
> > Thanks to the authors for the detailed feedback. My main concerns regarding the method's mechanics are resolved. However, given the billion-level parameter counts for both the encoder and decoder, the method currently serves as an interesting frontier exploration rather than a feasible solution for practical applications. As such, I will keep my score as is.

---

> > > ### Author Response · Authors · 2025-11-26
> > > **A feasible solution for practical applications**
> > >
> > > We sincerely thank the reviewer for the quick response and for confirming that the concerns regarding the method's mechanics have been resolved. We also appreciate the recognition of our work as an "interesting frontier exploration".
> > >
> > > We fully acknowledge the concern about the billion-level parameter counts. To demonstrate the potential for practical applications, we conduct further experiments using BLIP (a lightweight caption model) and consider a lightweight deployment strategy. We appreciate this valuable suggestion, and have included these experiments in Appendix A.8.
> > > * **Replacing BLIP-2 with BLIP**: We replaced the large BLIP-2 model with the more lightweight BLIP model. As shown in the BD-rate table below, utilizing BLIP (w/ LRE) yields compression performance very similar to the baseline BLIP-2.
> > > * **Lightweight deployment w/o LRE**: For scenarios strictly constrained by computational resources at the sender's end (e.g., mobile devices), we can directly transmit token indices without lossless re-encoding (LRE). This eliminates the need to deploy the VAR model at the sender, requiring only the VAE encoder and the lightweight caption model. Such strategy reduces the parameters at the sender's end from 2.2B to 268.8M. Although removing LRE results in a BD-rate increase ($\sim$ 25%) due to the lack of probability estimation, it offers a viable solution for edge devices.
> > >
> > >
> > > | Model | Sender Params | Receiver Params |
> > > | ------| ------------- | --------------- |
> > > | BLIP  | 223.9M| 0 |
> > > | VAR (w/o VAE) | 2.2B | 2.2B |
> > > | VAE  | 44.9M | 65.0M |
> > > |Total (w/ LRE) |2.2B |2.2B|
> > > |Total (w/o LRE)|268.8M |2.2B|
> > >
> > > > *Table 1: Model parameters for lightweight deployment using BLIP and w/o LRE.*
> > >
> > >
> > >
> > > | Method | BD-rate (%) - DISTS | BD-rate (%) - FID |
> > > | -------- | -------- | -------- |
> > > | BLIP-2 + w/ LRE| 0 |0 |
> > > | BLIP + w/ LRE| 0.12 | 0.08|
> > > | BLIP + w/o LRE | 25.93 | 25.56 |
> > >
> > > > *Table 2: BD-rate comparison for lightweight deployment using BLIP and w/o LRE. The BD-rate is calculated on DIV2K dataset with using BLIP-2 caption model and LRE as the baseline.*
> > >
> > > In summary, while ARPC explores the performance frontier, the results above confirm that the proposed framework can be a feasible solution for practical applications.

---

> > > > ### Comment · Reviewer_t8Sb · 2025-11-27
> > > >
> > > > Thanks for the clarifications. Since BLIP-2 brings limited performance improvement but requires significantly more parameters, BLIP is indeed a better choice. I recommend clarifying this point in the maintext of the paper.
> > > >
> > > > While the parameter count is still high compared to conventional learned image compression methods (even with BLIP and w/o LRE), I understand this is an inherent issue with generative compression and not a flaw specific to this work. Given the solidness of the proposed method, I maintain my original positive.

---

### Official Review · Reviewer_1Hec · 2025-10-28

**Soundness:** 4
**Presentation:** 2
**Contribution:** 3
**Rating:** 6
**Confidence:** 5

**Summary:**

This paper proposes an ultra-low bitrate image compression method based on the VAR model. Specifically, it introduces group-masked bitwise multi-scale residual quantization (GM-BMSRQ) and lossless re-encoding (LRE) techniques to improve compression ratio. The scale random dropout (SRD) strategy enhances the representation capability of earlier scales. The proposed ARPC method achieves superior perceptual and statistical fidelity on benchmark datasets.

**Strengths:**

This paper proposes an innovative extreme image compression method based on VAR. The manuscript is well-organized and original.

**Weaknesses:**

1.The Kodak dataset is a commonly used benchmark in image compression. However, the authors do not provide quantitative or qualitative comparisons of it. Please add these comparisons.
2.The paper does not compare the ARPC method with representative methods, such as token-based and one-step diffusion methods. The former includes the GLC [1] and DLF [2] methods, while the latter includes the RDEIC [3], OSCAR [4], and StableCodec [5] methods. Please add these comparisons for a comprehensive evaluation.
[1]Jia Z, Li J, Li B, et al. Generative latent coding for ultra-low bitrate image compression[C]. CVPR 2024.
[2]Xue N, Jia Z, Li J, et al. DLF: Extreme Image Compression with Dual-generative Latent Fusion[J]. ICCV 2025.
[3]Li Z, Zhou Y, Wei H, et al. RDEIC: Accelerating Diffusion-Based Extreme Image Compression with Relay Residual Diffusion[J]. TCSVT2025.
[4]Guo J, Ji Y, Chen Z, et al. OSCAR: One-Step Diffusion Codec Across Multiple Bit-rates[J]. NeurIPS2025.
[5]Zhang T, Luo X, Li L, et al. StableCodec: Taming One-Step Diffusion for Extreme Image Compression[J]. ICCV2025.
3.The related work lacks one-step, diffusion-based extreme image compression methods.
4.What is the bitwise multi-scale residual quantizer? Please explain how it works in detail.
5.Please explain how the scale random dropout strategy work and introduce it in detail.
6.Please explain the symbol d() in line 221 of the manuscript. Is it the MSE function?
7.For qualitative comparisons, the authors should select results for which the ARPC has the smallest BPP values compared to other competing methods. Please modify it (Fig. 4 and 5.).
8. Would you release the source code and pretrained models? We hope you can also release them to help us understand the ARPC.

**Questions:**

See the part of weaknesses.

---

> ### Author Response · Authors · 2025-11-23
> **Author response (part 1/2)**
>
> Dear Reviewer 1Hec, thank you very much for your careful review of our paper and insightful comments. We are encouraged by your positive comments on **innovative method**, **well-organized and original manuscript**. We hope the following responses can alleviate your concerns and clarify key points.
>
> ---
>
>
> **W1:** The Kodak dataset is a commonly used benchmark in image compression. However, the authors do not provide quantitative or qualitative comparisons of it. Please add these comparisons.
>
> **R1:** Thank you for the suggestion.  In response to your query, we provide the quantitative comparison on Kodak dataset in appendix of the revised version A.4. The results show that ARPC maintains outstanding performance on the Kodak dataset as well. And we provide the qualitative comparison in Figure 11.
>
>
> ---
>
> **W2:** The paper does not compare the ARPC method with representative methods, such as token-based and one-step diffusion methods.
>
> **R2:** Thank you for the suggestion. We are sorry for not considering these methods, and we further reproduce all the baselines you mentioned, including GLC [1], DLF [2], RDEIC [3], OSCAR [4], StableCodec [5], and ResULIC [6].  We update the results in Figure 3. The results show that ARPC can still outperform all the baselines on most metrics, especially FID and KID, demonstrating that ARPC can achieve superior perceptual and statistical fidelity. More importantly, compared to the above methods, **the most significant advantage of ARPC is the progressive coding**, which can not be achieved by the one-step diffusion based method.
> - **Compared with one-model-per-rate methods**: Such as StableCodec and DLF, one-model-per-rate methods need to train and store multiple different models for different bitrates. ARPC uses one model for continuous rate adaptation, significantly reducing the training and storage cost.
> - **Compared with multiple bitrates methods**: Such as OSCAR, these methods support variable bitrates, but generate an independent bitstream for each bitrate. In contrast, ARPC only generates a single bitstream and supports decoding from partial bitstreams, which is more flexible and more practical in bandwidth-constrained scenarios.
>
>
> ---
> **Reference**
>
> [1] Jia Z, Li J, Li B, et al. Generative latent coding for ultra-low bitrate image compression[C]. CVPR 2024.
>
> [2] Xue N, Jia Z, Li J, et al. DLF: Extreme Image Compression with Dual-generative Latent Fusion[J]. ICCV 2025.
>
> [3] Li Z, Zhou Y, Wei H, et al. RDEIC: Accelerating Diffusion-Based Extreme Image Compression with Relay Residual Diffusion[J]. TCSVT2025.
>
> [4] Guo J, Ji Y, Chen Z, et al. OSCAR: One-Step Diffusion Codec Across Multiple Bit-rates[J]. NeurIPS2025.
>
> [5] Zhang T, Luo X, Li L, et al. StableCodec: Taming One-Step Diffusion for Extreme Image Compression[J]. ICCV2025.
>
> [6] Ke A, Zhang X, Chen T, et al. Ultra Lowrate Image Compression with Semantic Residual Coding and Compression-aware Diffusion[J]. ICML2025.

---

> > ### Comment · Reviewer_1Hec · 2025-11-26
> >
> > The authors have addressed my concerns. Thank you.

---

> ### Author Response · Authors · 2025-11-23
> **Author response (part 2/2)**
>
> **W3:** The related work lacks one-step, diffusion-based extreme image compression methods.
>
> **R3:** We thank the reviewer for pointing this out.  We have revised the related works section to include recent one-step, diffusion-based extreme image compression methods, ensuring a more comprehensive overview of the state-of-the-art.
>
> ---
>
> **W4:** What is the bitwise multi-scale residual quantizer? Please explain how it works in detail.
>
> **R4:** We are sorry for confusing. The bitwise multi-scale residual quantizer (BMRSQ) is first proposed in [7]. For an feature map $F$, BMRSQ quantizes it into $K$ residual maps, denoted as $(R_1, R_2,...,R_K)$. The resolution of $R_k$, $(h_k \times w_k)$, grows larger gradually from $k=1,...,K$. Each scale corresponds to $h_k \times w_k$tokens, denoted as $r_{i,j}, i=1,...,h_k, j=1,...,w_k$. $r_{i,j}$ is a binary representation: $r_{i,j}=\frac{1}{\sqrt{c}}sign(\frac{r_{i,j}}{|r_{i,j}|})$, where $c$ is the dimension of latent space. We denote $F_k$ as the cumulative sum of the upsampled $R_{\leq k}$: $F_k = \sum_{i=1}^{k}upsample(R_i, (h,w))$. As $k$ increases, $F_k$ gradually approximates the continuous feature map $F$.
>
> ---
>
> **W5:** Please explain how the scale random dropout strategy work and introduce it in detail.
>
> **R5:** To enhance the representation capability of earlier scales, we use scale random dropout strategy. During training, with $K$ residual maps  $(R_1, R_2,...,R_K)$, we randomly drop out the last several residual maps with a ratio of $p$. With scale random dropout, the final output of BMSRQ is:
> $F\prime = \sum_{i=1}^{k}upsample(R_i, (h,w)), K_{start}\leq k \leq K$.
>
> ---
>
> **W6:** Please explain the symbol d() in line 221 of the manuscript. Is it the MSE function?
>
> **R6:** We are sorry for confusing. The symbol d denotes a general distortion metric to quantify the dissimilarity between the original image and the reconstructed image.
>
> ---
>
> **W7:** For qualitative comparisons, the authors should select results for which the ARPC has the smallest BPP values compared to other competing methods. Please modify it (Fig. 4 and 5.)
>
> **R7:** Thank you for the suggestion.  We have modified Figures 4 and 5.
>
> ---
> **W8:** Would you release the source code and pretrained models? We hope you can also release them to help us understand the ARPC
>
> **R8:** Yes, we will opensource upon acceptance.
>
> ---
> **Reference**
>
> [7] Han J, Liu J, Jiang Y, et al. Infinity: Scaling bitwise autoregressive modeling for high-resolution image synthesis[C]. CVPR2025.

---

> > ### Comment · Reviewer_1Hec · 2025-11-26
> >
> > Thank you for your detailed response. I have improved my score and hope that this paper will be accepted. Good luck to you!

---

> > > ### Author Response · Authors · 2025-11-26
> > > **Thanks to Reviewer 1Hec**
> > >
> > > Thank you for your response! We are grateful for your recognition of the innovation of our working methods and appreciate your efforts to enhance our work!

---

### Official Review · Reviewer_gRQE · 2025-10-29

**Soundness:** 3
**Presentation:** 3
**Contribution:** 2
**Rating:** 4
**Confidence:** 4

**Summary:**

This paper proposes ARPC (Autoregressive-based Progressive Coding) for ultra-low bitrate image compression. An image is quantized into K multi-scale residual token maps; the encoder transmits only the first k (coarse→fine), and a visual autoregressive (VAR) model predicts the untransmitted scales at the decoder (next-scale generation). ARPC also treats the VAR as a probability estimator for arithmetic (lossless) entropy coding of the transmitted tokens and introduces a group-masked bitwise multi-scale residual quantizer (GM-BMSRQ) to reduce bits at coarse scales.

**Strengths:**

- Dual use of VAR . Using VAR both to compress transmitted bits losslessly (arithmetic coding) and to generate untransmitted scales is technically neat and principled.
- Good progressive design. Encoding into hierarchical residual scales and stopping transmission at k aligns naturally with bitrate adaptation; the decoder’s next-scale VAR completes missing scales, yielding progressive reconstruction. The pipeline (lossless arithmetic decode → VAR generation → image decoder) is clearly illustrated.

**Weaknesses:**

- Complexity accounting. The claim of 2–6× faster decompression than diffusion is compelling, but a fuller wall-clock/compute-memory breakdown across image sizes and k values would strengthen the efficiency story.
- The impact analysis of missing text on reconstruction is lacking. The paper mentions using BLIP2 to extract image captions to assist reconstruction, but it does not analyze how the absence of text affects the final results, which is an incomplete approach.
- A more detailed baseline comparison is needed. Recently, numerous diffusion-based image compression methods have emerged, such as StableCodec[1] and ResULIC[2]. These methods have significantly improved decoding speed and performance, particularly StableCodec, which can complete image generation in a single step. This demonstrates that multi-step denoising is no longer a common limitation of diffusion-based methods. The authors need to conduct a more detailed performance and complexity comparison with these methods to highlight the significance of VAR-based approaches.

[1] Zhang, Tianyu, et al. "StableCodec: Taming One-Step Diffusion for Extreme Image Compression." arXiv preprint arXiv:2506.21977 (2025).

[2] A. Ke, X. Zhang, T. Chen, M. Lu, C. Zhou, J. Gu, and Z. Ma, “Ultra Lowrate Image Compression with Semantic Residual Coding and Compression-aware Diffusion,” in Proc. Int. Conf. Mach. Learn. (ICML), 2025.

**Questions:**

- In your introduction, I noticed the term "Infinite shared randomness." What exactly does this refer to? Is it about the impact of random seeds in diffusion models on the denoising process? This doesn’t seem to have a very significant impact.

---

> ### Author Response · Authors · 2025-11-23
> **Author response (part 1/2)**
>
> Dear Reviewer gRQE, thank you very much for your careful review of our paper and insightful comments. We are encouraged by your positive comments on **good progressive design**, **clear illustration of compression pipeline** and **neat and principled technical approach**. We hope the following responses can alleviate your concerns and clarify key points.
>
> ---
> **W1:** Complexity accounting. The claim of 2–6× faster decompression than diffusion is compelling, but a fuller wall-clock/compute-memory breakdown across image sizes and k values would strengthen the efficiency story.
>
> **R1:** Thank you for your suggestion. We have given the analysis of the impact of different transmitted scales k for encoding times in Appendix A.7.1. We also give the results below for an input image with a resolution of 1024 $\times$ 1024. Different from diffusion-based progressive codec, DiffC, which has consistent computational cost per iteration, the encoding complexity is low at low bitrates as fewer token counts at early scales. When transmitting tokens from fewer than 10 scales, the encoding time is under 3 seconds and the bitrate is below 0.025 bpp. As the number of tokens in the last 3 scales increases, it leads to a significant increase in compression time.
> | k | 1 | 2 | 3 | 4 | 5 | 6 | 7 | 8 | 9 | 10 | 11 | 12 | 13 |
> | --- | --- | --- | --- | --- | --- | --- | --- | --- | --- | --- | --- | --- | --- |
> | Encoding time (s) | 1.86 | 1.90 | 1.94 | 1.99 | 2.03 | 2.07 | 2.15 | 2.27 | 2.46 | 2.77 | 3.35 | 4.34 | 6.21 |
> | BPP | 7.3e-6 | 3.4e-5 | 1.4e-4 | 3.8e-4 | 9.6e-4 | 2.2e-3 | 4.6e-3 | 8.1e-3 | 0.013 | 0.025 | 0.042 | 0.064 | 0.108 |
>
> Furthermore, we provide the encoding and decoding times for different input image sizes. These experiments have also been added to Appendix A.7.1.
>
> | Image size | Encoding (s) | Decoding (s) |
> | --- | --- | --- |
> | 256 $\times$ 256 | 0.09 $\sim$ 0.45 | 0.44 |
> | 512 $\times$ 512 | 0.28 $\sim$ 1.39 | 1.33 |
> | 512 $\times$ 768 | 0.46 $\sim$ 2.40 | 2.29 |
> | 1024 $\times$ 1024 | 1.86 $\sim$ 6.21 | 5.39 |
>
>
> ---
>
> **W2:** The impact analysis of missing text on reconstruction is lacking. The paper mentions using BLIP2 to extract image captions to assist reconstruction, but it does not analyze how the absence of text affects the final results, which is an incomplete approach.
>
> **R2:** We have provided the ablation study of different text lengths and without text in Appendix A.5.3. As shown in Figure 14 in the manuscript, the results show that at ultra-low bitrates, the caption is crucial for semantic preservation. The tokens of earlier scales primarily encode structural layout, lacking finer semantic details of the original images, making the decompression more dependent on the guidance of the text. At higher bitrates, comparable performance can be achievable without the image caption.

---

> ### Author Response · Authors · 2025-11-23
> **Author response (part 2/2)**
>
> **W3:** A more detailed baseline comparison is needed. Recently, numerous diffusion-based image compression methods have emerged, such as StableCodec and ResULIC. These methods have significantly improved decoding speed and performance, particularly StableCodec, which can complete image generation in a single step. This demonstrates that multi-step denoising is no longer a common limitation of diffusion-based methods. The authors need to conduct a more detailed performance and complexity comparison with these methods to highlight the significance of VAR-based approaches.
>
> **R3:** Thank you for the suggestion. We further consider 6 more baselines, including GLC [1], DLF [2], RDEIC [3], OSCAR [4], StableCodec [5], and ResULIC [6]. For compression performance comparison, we update the results in Figure 3 of the revised manuscript. The results show that ARPC can still outperform all the baselines on most metrics, especially FID and KID, demonstrating that ARPC can achieve superior perceptual and statistical fidelity.
>
> Next, we give the complexity comparison. We choose the most representative diffusion based method, StableCodec, and ResULIC for comparison. We also report the BD-rate on CLIC2020 dataset for compression performance comparison, with ARPC as baseline, considering 3 metrics. The results are shown below.
>
>
>
> |Methods | Step| Encoding time (s) | Decoding time (s) | BD-rate(%)-DISTS | BD-rate(%)-FID | BD-rate(%)-PIEAPP |
> | --- | --- | ----------------- | ----------------- | ---------------- | -------------- | ----------------- |
> |PerCo | 20 |0.20 | 10.25 |1167.35 |882.09 |2744.75|
> |DiffEIC|50|0.65 | 15.98| 681.76 |139.64|100.75|
> |DiffPC|50 |0.17 | 23.66|93.90|20.49|16.52|
> |ResULIC|6|0.31|4.18|523.52|1469.67|761.27|
> |StableCodec|1|0.42|1.11|0.1547 |6.99 |254.42|
> | DiffC|-|3.63 $\sim$ 45.22 |13.63 $\sim$ 37.25|674.37|59.62 |117.11|
> |ARPC | 13 | 1.8 $\sim$ 6.2 | 5.39 | 0| 0| 0|
>
>
> Although ARPC requires more decoding time than one-step diffusion based methods, it achieves superior compression performance. More importantly, we highlight **the most significant advantage of ARPC is the progressive coding**, which can not be achieved by the one-step diffusion based method.
> - **Compared with one-model-per-rate methods**: Such as StableCodec and DLF, one-model-per-rate methods need to train and store multiple different models for different bitrates. ARPC uses one model for continuous rate adaptation, significantly reducing the training and storage cost.
> - **Compared with multiple bitrates methods**: Such as OSCAR, these methods support variable bitrates, but generate an independent bitstream for each bitrate. In contrast, ARPC only generates a single bitstream and supports decoding from partial bitstreams, which is more flexible and more practical in bandwidth-constrained scenarios.
>
>
>
> ---
>
> **Q1:** In your introduction, I noticed the term "Infinite shared randomness." What exactly does this refer to? Is it about the impact of random seeds in diffusion models on the denoising process? This doesn’t seem to have a very significant impact.
>
> **R1:** We are sorry for confusing. Infinite shared randomness is the core foundation for reverse channel coding (RCC), which is used by DiffC to achieve progressive coding. The sender and receiver utilize a prescribed shared seed to generate a completely identical sequence of pseudo-random numbers. The sender selects samples that match the target image distribution and transmits only the indices of these samples. The receiver uses the exact same random sequence to reconstruct the specific samples based on those indices. However, previous works have indicated that infinite shared randomness is not always available [7]. If the sender and receiver can't share randomness, the receiver is unable to decode correctly. As a result, DiffC is not practical for scenarios where infinite shared randomness is not available.
>
> ---
> **Reference**
>
> [1] Jia Z, Li J, Li B, et al. Generative latent coding for ultra-low bitrate image compression[C]. CVPR 2024.
>
> [2] Xue N, Jia Z, Li J, et al. DLF: Extreme Image Compression with Dual-generative Latent Fusion[J]. ICCV 2025.
>
> [3] Li Z, Zhou Y, Wei H, et al. RDEIC: Accelerating Diffusion-Based Extreme Image Compression with Relay Residual Diffusion[J]. TCSVT2025.
>
> [4] Guo J, Ji Y, Chen Z, et al. OSCAR: One-Step Diffusion Codec Across Multiple Bit-rates[J]. NeurIPS2025.
>
> [5] Zhang T, Luo X, Li L, et al. StableCodec: Taming One-Step Diffusion for Extreme Image Compression[J]. ICCV2025.
>
> [6] Ke A, Zhang X, Chen T, et al. Ultra Lowrate Image Compression with Semantic Residual Coding and Compression-aware Diffusion[J]. ICML2025.
>
> [7] Lei E, Hassani H, Bidokhti S S. Optimal neural compressors for the rate-distortion-perception tradeoff[J].NeurIPS2025.

---

> ### Author Response · Authors · 2025-11-27
> **A Gentle Reminder of the Final Feedback**
>
> Please allow us to thank you again for reviewing our paper and the insightful comments, and in particular for recognizing the strengths of our paper in terms of good progressive design, neat and principled technical approach, good soundness, and good presentation.
>
> Kindly let us know if our response and the new experiments have properly addressed your concerns. We are more than happy to answer any additional questions during the post-rebuttal period. Your feedback will be greatly appreciated.

---

### Official Review · Reviewer_joRB · 2025-11-17

**Soundness:** 3
**Presentation:** 3
**Contribution:** 2
**Rating:** 4
**Confidence:** 4

**Summary:**

This paper introduces ARPC (AutoRegressive-based Progressive Coding), which leverages a Visual Autoregressive Model (VAR) based on next-scale prediction to for image compression. The VAR also serves as a probability estimator for near-lossless entropy coding. To further improve compression efficiency and semantic representation, the authors propose a Group-Masked Bitwise Multi-Scale Residual Quantizer (GM-BMSRQ) and a Scale Random Dropout (SRD) strategy. Experiments on DIV2K-val and CLIC2020 demonstrate that ARPC surpasses both diffusion-based and VQ/GAN-based baselines in perceptual metrics.

**Strengths:**

By transmitting only the first k scales and autoregressively completing the rest with VAR, it enables progressive transmission and adaptive bitrate control. The VAR module serves both as a generator and a probability estimator, reducing overall bitrate while supporting lossy and lossless compression. GM-BMSRO further enhances the bitwise multi-scale residual quantizer by group masking to early scales, while SRD encourages these scales to capture richer semantic information. Extensive evaluations across diverse datasets and metrics demonstrate the robustness and practical effectiveness of these techniques.

**Weaknesses:**

(1) The structural diagram (Figure 2) fails to explicitly show the image caption’s presentation form and functional mechanism. While the text states captions (generated via BLIP2) act as global semantic context to guide VAR’s autoregressive prediction of unreceived scales, Figure 2 lacks labels for the caption generation module.
(2) The grouping logic and parameter settings of GM-BMSRQ lack sufficient justification. The paper proposes dividing the K scales into three groups and masking the last c/2 channels of the first group and the last c/4 channels of the second group to reduce bit cost. However, the rationale behind these key design choices is not explained: (i) the basis for selecting the specific channel masking numbers (e.g., c/2) is unclear, as no experiments compare different masking configurations; (ii) the logic for dividing scales into three groups rather than two or four, and the criteria for allocating scales to each group, is not discussed; (iii) the effect of different channel configurations (e.g., 8, 12, 16 channels) on compression performance is only partially explored through comparisons of c=32 and c=16, leaving the module’s design insufficiently validated and its optimality unproven.
(3) The paper exhibits notable deficiencies in evaluating decoding efficiency and comparing with baseline methods. On one hand, although the ARPC decoding time is reported as 5.39s and claimed to be 2–6× faster than diffusion-based methods, the breakdown of this runtime is not provided—key steps such as VAR autoregressive prediction, arithmetic decoding, and image reconstruction are not individually profiled, making it difficult to identify efficiency bottlenecks and guide further optimization. More importantly, this decoding time is not compared against the current state-of-the-art in ultra-low bitrate image compression. On the other hand, the selection of baseline methods is limited and does not include recent mainstream approaches. To fully validate ARPC’s performance, comparisons should be extended to methods such as DLF[1] (Extreme Image Compression with Dual-generative Latent Fusion), GLC[2], and single-step diffusion methods like StableCodec[3], thereby providing a more comprehensive evaluation and clarifying ARPC’s position relative to the current research frontier.

[1] N. Xue, Z. Jia, J. Li, B. Li, Y. Zhang, and Y. Lu, “DLF: Extreme Image Compression with Dual-generative Latent Fusion,” ICCV, 2025.
[2] Jia, Zhaoyang, et al. "Generative latent coding for ultra-low bitrate image compression." CPVR, 2024.
[3] Zhang, Tianyu, et al. "StableCodec: Taming One-Step Diffusion for Extreme Image Compression." ICCV, 2025.

**Questions:**

(1)	Are the bits used for text encoding included in the reported bpp statistics? How much bitrate do the captions contribute to the total bpp?
(2)	The paper states that scale random dropout (SRD) is applied with a probability of 0.2 from the fourth scale during training—why is the fourth scale chosen as the starting point for SRD, and how does adjusting the dropout probability (e.g., 0.1 or 0.3) affect the model’s ability to preserve semantic information in earlier scales?
(3)	For images with complex textures (e.g., dense text or fine-grained patterns), does ARPC require adjusting hyperparameters (e.g., number of scales K, channel dimensions of GM-BMSRQ groups) to maintain reconstruction quality, and if so, what guidelines exist for such adjustments?

---

> ### Author Response · Authors · 2025-11-17
> **So, you just have no shame?**
>
> So you finally realize that **copying other reviewer's comments** is not good?
>
> And then you **use LLM to generate another version of comments** to pretend nothing happened?

---

### Author Response · Authors · 2025-11-17
**Extremely irresponsible reviewer joRB (copy other comments first and then replace with an AIGC version)**

Dear AC, reviewers, and readers from the community,

Please let us rephrase what happened:
1. When review release, we found the **comments from reviewer joRB and gRQE are EXACTLY THE SAME**.
2. We did not report in the beginning but then, it is ridiculous that **reviewer joRB deleted the repeated comments and upload an AIGC version today** to pretend nothing just happened.

Evidence see: https://anonymous.4open.science/r/Submission6729-4834/Readme.md

We have no idea how this could happen. I used to try to find some reviewer's policy or guideline to prove this behavior violated something. But then I realized that **this is way too much**. Never heard such an operation (**copy comments and then replace with an AIGC one**) in the past many years.

We have **only one appeal** as follows.
**Please mark reviewer joRB as irresponsible reviewer and desk reject reviewer joRB's submission (refers to previous year's CVPR experience) and further consider to ban reviewer joRB for following venues.**

In the end, we know if we deal with this privately maybe our paper has a higher chance to be accepted (e.g. report to AC with humble message like please ignore this review...). But I’m really at the end of my rope with this so I have decided to make this open in this OpenReview system.

On behalf of Authors of submission 6729.

---

> ### Comment · Area_Chair_Fxhv · 2025-11-17
>
> Dear Authors,
>
> Thank you for raising your concerns.
>
> - Reviews are advisory tools for the decision process. I take the duplication of comments seriously and will factor this into my assessment and final recommendation.
>
> - If an initial review was added or changed after the deadline, it’s up to you whether to reply. You don’t have to respond to those updates.
>
> - Concerns about review independence or the use of generative tools will be handled with the Senior Area Chairs/Program Chairs in accordance with ICLR policy.
>
> Thank you for your understanding and professionalism.
>
> AC

---

### Author Response · Authors · 2025-11-23
**General response**

We thank the reviewers for their insightful comments and acknowledging that the paper is **innovative** (1Hec), **well-organized** (1Hec/t8Sb), has **effective progressive design with VAR** (gRQE/t8Sb), **clear illustration of compression pipeline** (gRQE), **neat and principled technical approach** (gRQE). We have carefully considered your comments and will take them into account to improve our work. Before we respond to each reviewer individually, we address common concerns as follows.

**W1:** More baselines for comparison.

**R1:** According to the suggestions of reviewers gRQE and 1Hec, we further consider 6 baselines, including GLC [1], DLF [2], RDEIC [3], OSCAR [4], StableCodec [5], and ResULIC [6]. The results show that ARPC can still outperform all the baselines on most metrics, especially FID and KID, demonstrating that ARPC can achieve superior perceptual and statistical fidelity. More importantly, compared to the above methods, **the most significant advantage of ARPC is the progressive coding**, which can not be achieved by the one-step diffusion based method.

- **Compared with one-model-per-rate methods**: Such as StableCodec and DLF, one-model-per-rate methods need to train and store multiple different models for different bitrates. ARPC uses one model for continuous rate adaptation, significantly reducing the training and storage cost.
- **Compared with multiple bitrates methods**: Such as OSCAR, these methods support variable bitrates, but generate an independent bitstream for each bitrate. In contrast, ARPC only generates a single bitstream and supports decoding from partial bitstreams, which is more flexible and more practical in bandwidth-constrained scenarios.

---

Next, we summarize the main concerns raised by the reviewers and highlight the corresponding enhancements we made.

* **Additional baselines (Reviewer gRQE & 1Hec).**
    * In Section 4.2, we add 6 new baselines for quantitative comparisons (updated Figure 3) and two representative one-step diffusion based methods, StableCodec and ResULIC, for computational complexity comparisons (updated Table 1).
* **Analysis of detailed computational complexity (Reviewer gRQE).**
    * In Appendix A.7.1, we add more detailed results of encoding and decoding time with different $k$ and input image sizes to strengthen the efficiency analysis (updated Tables 5 and 6).
* **Analysis of missing text (Reviewer gRQE).**
    * We note that this experiment is already covered in Appendix A.5.3 and give analysis about how the absence of text affects the final results.
* **Additional dataset evaluation (Reviewer 1Hec).**
    * In Appendix A.4, we provide a new evaluation on Kodak dataset, including quantitative and qualitative comparisons, to demonstrate ARPC maintains outstanding performance on the Kodak dataset as well (updated Figures 10 and 11).
* **Related works update (Reviewer 1Hec).**
    *  We have revised the related works section to include recent one-step, diffusion-based extreme image compression methods.
*  **Qualitative results modification (Reviewer 1Hec).**
    *  Figures 4 and 5 have been revised to ensure ARPC has the smallest BPP.
*  **Clarification of encoding time and model parameters (Reviewer t8Sb).**
    * We clarify that **the time increase with bitrate is an inherent characteristic of progressive coding rather than a design flaw**.
    * We highlight that **utilizing text as a global semantic prior has become a standard paradigm** in diffusion-based compression.
    * We have reported detailed parameter counts for both the encoder and decoder to address the concern regarding model weights.
* **Validation of improvements over vanilla VAR (Reviewer t8Sb).**
    * We clarified that directly adapting vanilla VAR is suboptimal and **ARPC effectively bridges the gap between generation and compression**.
    * We also compare ARPC with vanilla VAR to demonstrate the improvement of ARPC in Appendix A.5.5 (updated Table 4).
* **Ablation on specific mask choices (Reviewer t8Sb).**
    * In Appendix A.5.4, we include ablation experiments on different specific mask choices for the Group-Masked (GM-BMSRQ) method to verify that our default choice achieves the optimal trade-off (updated Table 3).
---
**Reference**

[1] Jia Z, Li J, Li B, et al. Generative latent coding for ultra-low bitrate image compression[C]. CVPR 2024.

[2] Xue N, Jia Z, Li J, et al. DLF: Extreme Image Compression with Dual-generative Latent Fusion[J]. ICCV 2025.

[3] Li Z, Zhou Y, Wei H, et al. RDEIC: Accelerating Diffusion-Based Extreme Image Compression with Relay Residual Diffusion[J]. TCSVT2025.

[4] Guo J, Ji Y, Chen Z, et al. OSCAR: One-Step Diffusion Codec Across Multiple Bit-rates[J]. NeurIPS2025.

[5] Zhang T, Luo X, Li L, et al. StableCodec: Taming One-Step Diffusion for Extreme Image Compression[J]. ICCV2025.

[6] Ke A, Zhang X, Chen T, et al. Ultra Lowrate Image Compression with Semantic Residual Coding and Compression-aware Diffusion[J]. ICML2025.

---

### Comment · Area_Chair_Fxhv · 2025-11-27

Dear Reviewers,

Author responses are now posted. Please add your discussion comment(s) and update score/confidence as needed. Thank you!

Best regards,

AC

---

### Meta-Review · Area_Chair_hWxJ · 2026-01-10

**Summary:**

[AC: The authors raised the issue of receiving duplicated reviews in the initial review round. They indicated that “the initial comments from Reviewer gRQE and Reviewer joRB were exactly the same. Additionally, Reviewer joRB deleted the repeated comments and uploaded an AIGC version after the rebuttal started.” I believe all these reviews should be disregarded and recommend the TPC take proper measures to look into this case.]

[AC: I however find the comments from gRQE valid. The authors have also addressed them accordingly.]

-	Reviewer gRQE (4: marginally below the acceptance threshold. But would not mind if paper is accepted; 4: You are confident in your assessment.)

o	Complexity accounting. The claim of 2–6× faster decompression than diffusion is compelling, but a fuller wall-clock/compute-memory breakdown across image sizes and k values would strengthen the efficiency story.

o	A more detailed baseline comparison is needed. Recently, numerous diffusion-based image compression methods have emerged, such as StableCodec[1] and ResULIC[2]. These methods have significantly improved decoding speed and performance, particularly StableCodec, which can complete image generation in a single step. This demonstrates that multi-step denoising is no longer a common limitation of diffusion-based methods. The authors need to conduct a more detailed performance and complexity comparison with these methods to highlight the significance of VAR-based approaches.

[AC: Decoding runtime results were previously provided in the Appendix. The authors also additionally updated Table 1 to report runtime comparisons with StableCodec and RDEIC, showing promising results. But, it is unclear whether all the baseline methods are run on the same machine. Runtimes can vary significantly depending on the underlying compute platform. Moreover, compute-memory breakdown analyses, such as MAC/pixel and other platform independent metrics, are requested but not provided.]

[AC: The authors emphasized that progressive/scalable coding is a notable feature of the proposed method, which I agree with and is a big plus in the proposed method.]

o	The impact analysis of missing text on reconstruction is lacking. The paper mentions using BLIP2 to extract image captions to assist reconstruction, but it does not analyze how the absence of text affects the final results, which is an incomplete approach.

[AC: Quantitative results have been provided in Appendix. A.5.3 to clarify the effectiveness of the image caption. The authors claim that the caption is crucial for semantic preservation at low rates and not so critical at high rates. However, the claim would have been better supported by providing additional sample images, particularly for low k values, such as k=1,2,3. I think the current rebuttal response is not satisfactory and convincing enough.]

-	Reviewer 1Hec (6: marginally above the acceptance threshold. But would not mind if paper is rejected; 5: You are absolutely certain about your assessment.)

o	The Kodak dataset is a commonly used benchmark in image compression. However, the authors do not provide quantitative or qualitative comparisons of it. Please add these comparisons.

[AC: Results were additionally generated during the rebuttal period and provided in A.4. The results appear to be promising.]

o	The paper does not compare the ARPC method with representative methods, such as token-based and one-step diffusion methods. The former includes the GLC [1] and DLF [2] methods, while the latter includes the RDEIC [3], OSCAR [4], and StableCodec [5] methods. Please add these comparisons for a comprehensive evaluation.

[AC: Results were additionally generated during the rebuttal period and provided in Fig. 3. A similar request was made by another reviewer. The results appear to be promising. Moreover, the proposed method has the additional benefit of being able to do progressive/scalable coding.]

[AC: The other questions are mostly clarification questions and have been addressed. The reviewer had read the rebuttal letter and was okay with the revisions.]

o	The related work lacks one-step, diffusion-based extreme image compression methods.
o	What is the bitwise multi-scale residual quantizer? Please explain how it works in detail.
o	Please explain how the scale random dropout strategy work and introduce it in detail.
o	Please explain the symbol d() in line 221 of the manuscript. Is it the MSE function?
o	For qualitative comparisons, the authors should select results for which the ARPC has the smallest BPP values compared to other competing methods. Please modify it (Fig. 4 and 5.).
o	Would you release the source code and pretrained models? We hope you can also release them to help us understand the ARPC.

-	Reviewer T8Sb (6: marginally above the acceptance threshold. But would not mind if paper is rejected; 5: You are absolutely certain about your assessment.)

o	The encoding time increases with the bitrate. Additionally, the encoding process requires a caption model to generate captions, making the encoder relatively "heavy." Could the authors provide the parameter counts for the models used at both the encoder and decoder ends?

[AC: Additional results were provided. From the results, the proposed method is advantageous as compared to the baseline methods. In addition, the authors provided an ablation study in A.5.3 that provides further insights into how the image caption may help improve coding efficiency particularly at low rates.]

o	The ablation study in Figure 7 shows that while each proposed module contributes some improvement, the individual gains are relatively modest. Could it be that directly using VAR for encoding and decoding already offers a performance advantage over other generative compression methods?

[AC: Additional results were provided, confirming that using VAR directly is not able to achieve the performance reported in the paper. Changes were also made to the training process and entropy coding. This major concern is addressed properly.]

o	Further experiments are needed to investigate the impact of different specific mask choices for the Group-Masked (GM-BMSRQ) method.

[AC: Additional results have been added. This is not the most critical concern from the reviewer.]

**Reviewer Concerns:**

See my comments in the summary section.

**Reviewer Scores:**

The reviews were mostly positive: 6, 6, 4. Reviewer gRQE may stick to his initial rating 4 because his comments may not be fully addressed. However, I found that most other comments (from gRQE and the other two reviewers) are addressed by providing concrete data / evidence to justify the superiority of the proposed method. It is noted that the proposed method may still be far from practical due to the huge model size. However, this aspect is not specific to this work and is common for most generation-based compression. Given the pros and cons, I would recommend accepting it.

---

### Decision · Program_Chairs · 2026-01-26

Accept (Poster)